

# Impacts of land-use history on the recovery of ecosystems after agricultural abandonment

Andreas Krause[1], Thomas A. M. Pugh[1,2], Anita D. Bayer[1], Mats Lindeskog[3], Almut Arneth[1]

[1]Karlsruhe Institute of Technology, Institute of Meteorology and Climate Research – Atmospheric Environmental Research (IMK-IFU), Kreuzeckbahnstr. 19, 82467 Garmisch-Partenkirchen, Germany
[2]School of Geography, Earth and Environmental Science, University of Birmingham, B15 2TT, United Kingdom
[3]Department of Physical Geography and Ecosystem Science, Lund University, 22362 Lund, Sweden

*Correspondence to*: Andreas Krause (andreas.krause@kit.edu)

**Abstract.** Land-use changes have been shown to have large effects on climate and biogeochemical cycles, but so far most studies have focused on the effects of conversion of natural vegetation to croplands and pastures. By contrast, relatively little is known about the influence of past agriculture on vegetation regrowth and carbon sequestration following land abandonment, decades or even centuries later. We used the LPJ-GUESS dynamic vegetation model to study the legacy effects of different land-use histories (in terms of type and duration) across a range of ecosystems. To this end, we performed six idealized simulations for Europe and Africa in which we made a transition from natural vegetation to either pasture or cropland, followed by a transition back to natural vegetation after 20, 60 or 100 years. The simulations identified substantial differences in recovery trajectories of four key variables (vegetation composition, vegetation carbon, soil carbon, Net Biome Productivity) after agricultural cessation. Vegetation carbon and composition typically recovered faster than soil carbon in sub-tropical, temperate and boreal regions, and vice versa in the tropics. While the effects of different land-use histories on recovery periods of soil carbon stocks often differed by centuries across our simulations, differences in recovery times across simulations were typically small for Net Biome Productivity (a few decades) and modest for vegetation carbon and composition (several decades). Spatially, we found the greatest sensitivity of recovery times to prior land-use in boreal forests and subtropical grasslands where post-agricultural productivity was strongly affected by prior land management. Our results suggest that land-use history is a relevant factor affecting ecosystems long after agricultural cessation and it should be considered not only when assessing historical or future changes in simulations of the terrestrial carbon cycle, but also when establishing long-term monitoring networks and interpreting data derived therefrom, including analysis of a broad range of ecosystem properties or local climate effects related to land cover changes.

Keywords: land-use history, agricultural abandonment, legacy effects, ecosystem recovery, carbon sequestration



# 1    Introduction

Historically, many natural forests or grasslands on Earth have been cleared or cultivated for grazing, timber, food production, mining or settlements. However, land-use change (LUC) in these areas has rarely been continuous, and land cover and management have often changed for a variety of reasons (Burgi and Turner, 2002). While currently about 39 % of the ice-free land is actively used by humans for agriculture or urban areas, it has been estimated that a similar area is influenced by some form of past human usage, for instance agricultural abandonment or changing forest management (NatureEditorial, 2009), even though many of these areas may look undisturbed at first glance. Recently, agricultural cessation rates have risen globally, especially in the temperate region. For example, during the last decades, large areas in Europe previously used for pasture or crop cultivation were abandoned (e.g. Schierhorn et al., 2013; Smith et al., 2005). Following agricultural abandonment, and in the absence of further anthropogenic influence, natural vegetation recolonizes in a typical succession from herbaceous vegetation to shrubland and forests, if environmental conditions are suitable for tree growth. These secondary forests act as an important carbon (C) sink during the years of regrowth, thereby significantly reducing the growth rate of global atmospheric $CO_2$ concentration (Pan et al., 2011).

The immediate effects of land-use (LU) practices on C fluxes and nutrient cycles have been studied in some detail over recent decades. Generally, agriculture significantly reduces C and, in the absence of supplementary sources, nitrogen (N) pools due to initial deforestation, reduced soil litter input, and accelerated soil decomposition and erosion (Davidson and Ackerman, 1993; Fujisaki et al., 2015; Guo and Gifford, 2002; McLauchlan, 2006; Murty et al., 2002). Pasture soils can be an exception, as they have been found to accumulate C, depending on location and management (McSherry and Ritchie, 2013; Milchunas and Lauenroth, 1993). The long-term importance of past LU on ecosystems, however, was recognized only recently, and much less effort has been put so far into the investigation of legacy effects of LU history on ecosystem processes, how long these effects persist, or if they may even be irreversible (Chazdon, 2014; Compton and Boone, 2000; Cramer et al., 2008; Hobbs et al., 2009; McLauchlan, 2006). This is not only important to understand present-day ecological systems, but also because, due to demographical, social, technological, economic and environmental changes, LUC and land abandonment will continue to occur in the future (Hurtt et al., 2011).

Most observational studies that looked at the recovery of ecosystems after agricultural cessation focused on the first years of succession. Analyses of the long-term effects of historical LU are often limited by the unavailability of adequate LU information, or the absence of undisturbed ecosystems, and usually rely on chronosequences (Chazdon, 2003; Knops and Tilman, 2000). Only a few long-term observational study plots like the one maintained at the Rothamsted Experimental Station (e.g. Poulton et al., 2003) exist. Differences between (near) pristine and post-agricultural forests or grasslands have been reported to persist for decades or centuries after agricultural abandonment for various variables, including soil pH (Falkengren-Grerup et al., 2006), microbial communities (Fichtner et al., 2014), soil C, N and phosphorus (Compton and Boone, 2000), and other nutrients (Wall and Hytonen, 2005). Furthermore, above-ground biomass (Wandelli and Fearnside,



2015), percentage vegetation cover (Lesschen et al., 2008), biodiversity (Vellend, 2004), species composition (Aide et al., 2000) and structure (Bellemare et al., 2002) remained affected for years to decades, or even longer. These effects have consequences, not only for C sink capacity of the ecosystem, but also for water and energy exchange between the land and the atmosphere (Foley et al., 2003), which also has important, albeit still highly uncertain, implications for regional climate

change (e.g. Arora and Montenegro, 2011; Brovkin et al., 2013; de Noblet-Ducoudre et al., 2012). Some studies detected an influence of ancient agriculture on forest composition and diversity even thousands of years later (Dambrine et al., 2007; Dupouey et al., 2002; Willis et al., 2004). However, the persistence of legacy effects varies considerably with former LU, geographical location, sampling methods and examined variables, making recovery trajectories often hard to predict (Cramer et al., 2008; Foster et al., 2003; Guariguata and Ostertag, 2001; Post and Kwon, 2000; Suding et al., 2004).

In this study, we performed stylized simulations with the LPJ-GUESS dynamic global vegetation model (DGVM) to explore the importance of agricultural LU history in terms of type and duration for the regeneration of ecosystems and C stocks and fluxes under a range of environmental conditions. We converted natural vegetation to either pasture or cropland, followed by a re-transition to natural vegetation after a varying time period (20, 60, 100 years). While there are numerous variables suitable to measure recovery (Chazdon, 2003), we analyzed recovery times for vegetation composition (represented here by

the dominant Plant Functional Type, in terms of leaf area index), vegetation C, soil C, and Net Biome Productivity, to evaluate the longevity of the effects of LU history on the C cycle component of ecosystems, and to ascertain if the system eventually recovers to its pre-disturbance state.

## 2      Methods

### 2.1      LPJ-GUESS

LPJ-GUESS is a process-based DGVM that is driven by climate, atmospheric $CO_2$ concentration and N input (Smith et al., 2014). Plants are attributed to one of 11 Plant Functional Types (PFTs, 9 groups of tree species and two grasses) which are distinguished, for instance, in terms of their climate preferences for establishment and survival, photosynthetic pathways, growth rates, and growth strategies. Vegetation dynamics and composition at a given location result from competition

between plants for light and soil resources in a number of independent replicate patches (50 in this study), averaged per 0.5° x 0.5° grid-cell. Wildfire is included in the model and additionally, stochastic disturbances kill all the biomass in a patch, representing e.g. storm or insect damages, with a typical return period of 100 years (Smith et al., 2014). Recent model updates comprise the representation of LUC (Lindeskog et al., 2013) and the implementation of the N cycle in natural vegetation and grasses (Smith et al., 2014). The representation of the N cycle is crucial for this study because previous

agricultural N dynamics, such as extraction through harvest and input through fertilization, can greatly affect ecosystems even after many decades (Richter et al., 2000).



Conversion of natural to managed land in LPJ-GUESS is characterized by the initial killing of all living vegetation in the affected area. The corresponding woody biomass is partly oxidized immediately (67-76 %) and partly transferred to the product (21 %) or litter (3-12 %) pool. 10 % of the leaves are oxidized while the rest of the leaves and the fine roots enter the litter pool. Only the litter thus remains in the ecosystem post land-conversion. Pastures are represented by preventing tree establishment and wildfires and by splitting the above-ground biomass of the grasses equally between atmosphere (harvest) and litter at the end of each year. Crops were represented by grass PFTs modified to mimic aspects of cropland important for the C and N cycles. Settings for croplands and pastures were as follows:

1. For transitions from natural vegetation to cropland, we transferred only 3 % of the cleared woody biomass to the litter instead of 12 % for natural vegetation – pasture transitions. This accounts for the practice that farmers would try to remove as many coarse roots as possible before planting of crops.

2. Harvest efficiency (in this study: fraction of above-ground biomass that is oxidized) was 0.5 yr$^{-1}$ for pasture, representing the net effect of grazing processes (Lindeskog et al., 2013). For crop simulations we changed the harvest efficiency to 0.8 yr$^{-1}$, representing simplified crop harvest, as in Lindeskog et al. (2013).

3. While we removed 100 % of harvested N biomass for croplands we changed this value to 65 % for pastures. That accounts for significant urine N regain from animals fed on pastures (Dean et al., 1975; Lauenroth, 1992).

4. Root turnover rate was 0.7 yr$^{-1}$ for pasture and was adapted to 1.0 yr$^{-1}$ for croplands to represent the annual plant types used in most croplands.

5. In croplands we estimated tillage effects by increasing heterotrophic respiration by a factor of 1.94 (Pugh et al., 2015).

6. We simulated N fertilization in croplands by applying 75 kg ha$^{-1}$ yr$^{-1}$ equally throughout the year.

After patch-destroying disturbances or managed land converting back to natural vegetation, there is a typical succession from grasses to light-demanding pioneer trees, eventually followed in many ecosystems by the establishment of shade-tolerant species. It has been shown that LPJ-GUESS is able to realistically simulate observed succession pathways and species variations (Hickler et al., 2004; Smith et al., 2014).

## 2.2 Simulation setup

During spin-up (500 years) and the simulation period (900 years), we forced LPJ-GUESS with temperature-detrended, repeated 1981-2000 climate from the University of East Anglia Climate Research Unit 3.0 dataset (CRU, 2013), 1990s mean N deposition (Lamarque et al., 2013) and a fixed atmospheric $CO_2$ mixing ratio of 356 ppmv. We ran the model for Europe and Africa (33°E to 55°W), covering a wide range of environmental conditions, including examples of all major biomes (Smith et al., 2014). We chose Africa and Europe for the simulation domain because the original LU version of the model was evaluated against observations in Africa (Lindeskog et al., 2013) and to limit the computational expense of the



simulations. We did not intend to realistically represent typical crop and pasture management across the domain (i.e. the spatial variability in fertilizer use, multiple cropping systems, or irrigation). For all simulations we used potential natural vegetation cover to spin-up the model, followed by a transition to either pasture or croplands directly after spin-up and a transition back to natural vegetation after a varying time period of 20, 60 and 100 years. This resulted in three pasture (P20, P60, P100) and three cropland (C20, C60, C100) simulations. Additionally, we performed a reference simulation in which natural vegetation was retained throughout the whole simulation period.

## 2.3 Analyzed grid-cells and biome classification

To facilitate the interpretation, we classified each grid-cell to one biome. We used the same classification rules as Smith et al. (2014), aggregated to 8 biomes as in Bayer et al. (2015). Afterwards, we excluded grid-cells from the analyses which were classified as desert or tundra, had a mean NPP below 0.1 kg C m$^{-2}$ yr$^{-1}$, or were located above 62.5°N, making the assumption that the relevance of these low-productive areas for agriculture is negligible.

## 2.4 Analyzed variables and definition of recovery

We studied the influence of LU history on ecosystems by analyzing four key variables: dominant PFT (in terms of leaf area index, LAI), vegetation C, soil C (excluding litter) and the sum of all ecosystem-level C fluxes, termed Net Biome Productivity (NBP). We investigated the legacy effects of LU history by calculating a recovery time for each variable, simulation and grid-cell after the conversion back to natural vegetation. For vegetation C, soil C and NBP recovery time was defined as the year in which the 20-year running mean of the variable exceeded the threshold of one standard deviation (σ) below the mean of the reference simulation for the first time after agricultural abandonment (precondition 1). σ was also calculated on the 20-year running mean of the reference simulation. To avoid "false positive" identifications of recovery in cases for which the variable of interest was initially within 1 σ, but then exhibited dynamics taking it outside this range (e.g. soil C in Fig. A1), we applied an additional criterion of whether the minimum after the transition to natural vegetation was located in the first 200 years and if it was below the mean minus 1 σ threshold. If that was the case, the precondition was expanded so that recovery was only possible after the year in which the minimum occurred. Choosing a 200 year window seems reasonable because the minimum occurred within the first 200 years for all biome averages of all variables and simulations. If the minimum was located after 200 years, we assumed the minimum to be a result of natural variability and recovery was considered to be achieved as soon as the variable in question exceeded the threshold of 1 σ below the mean. Figure A1 shows an example of how soil C recovery was calculated for one site. For the dominant PFT, we used the same definition as for vegetation C, soil C and NBP but further checked if the dominant PFT was the same as in the reference simulation (precondition 2). For all variables, the recovery time was capped at 800 years after reconversion to natural vegetation when recovery was not achieved at the end of the simulation period, implying a lower limit for these cases.



## 3    Results

### 3.1    Reference simulation

Maps of vegetation and soil C, as well as dominant PFT and biomes derived from PFT composition for the reference simulation are shown in Fig. 1. The salient features of biome and C storage distribution at the regional scale are captured (Haxeltine and Prentice, 1996; Scharlemann et al., 2014). Vegetation C reaches its highest values in tropical forests of central Africa and decreases towards the deserts of southern and northern Africa. Patterns are more homogeneous in Europe where most areas store 5-10 kg C m$^{-2}$. Similar to vegetation C, soil C in the (sub)tropics also decreases with drier conditions, however, the differences are small with typical values of 5-10 kg C m$^{-2}$. Soils in the temperate and southern boreal ecosystems of Europe generally store more C (usually >10 kg C m$^{-2}$), especially in colder environments. While Europe is mostly dominated by woody PFTs (e.g. TeBS is the acronym for temperate broadleaved summergreen tree), in Africa there is a shift from C3 and C4 grasses in the dry regions to trees in the humid tropics. This gradient also appears in the corresponding biome map: In Africa and the Arabian Peninsula, LPJ-GUESS reproduces the transition from grasslands to savannas and tropical forests (TrFo) as the equator is approached. Europe is mostly classified as temperate forests (TeFo), with some boreal forests (BoFo) in the north and some shrublands/savannas in the south.

### 3.2    Dominant PFT recovery

The LAI of the dominant PFT recovers on average within around one century for all LU histories (Fig. 2). Maps of the recovery time (Fig. 3) show distinct geographical patterns which occur in all simulations. Most subtropical grasslands and savannas, and parts of the temperate and boreal forests recover within several decades, some grasslands even within five years. Contrary, in other parts of the temperate forests and in the tropical forests, recovery times are clearly longer (>100 years). Long recovery is associated with woody successional vegetation dynamics, as slow recovering areas are usually dominated by temperate broadleaved summergreen and tropical broadleaved evergreen forests (compare PFT distribution in Fig. 1). These are shade-tolerant PFTs that establish only slowly after disturbances. For 84 % of all analysed grid-cells, precondition 1 (LAI recovery) was the delaying precondition for dominant PFT recovery (numbers exemplified for the P60 simulation), compared to only 3 % for precondition 2 (dominance recovery). For the remaining grid-cells, both preconditions were fulfilled in the same year.

Overall, differences across simulations of different LU histories are moderate, with generally only small differences in temperate forests, savannas and shrublands (Fig. 3, see also biome averages in Table 1 and the histogram in Fig. A2). Areas of major differences are Central Africa where P20 recovers faster than other simulations because post-agricultural net mineralization rates are higher in this region for P20 than for the other simulations (Fig. 4), thereby relatively increasing post-agricultural N availability compared to the other simulations (Fig. 5), and the African Mediterranean coast where croplands recover much faster because the reduced C:N ratio in the soil (not shown) enhances N mineralization and thus



plant N availability compared to pastures. Furthermore, in parts of the boreal zone recovery takes several hundred years for C100 instead of a few decades for the other simulations. Figure 6 shows the maximum differences in recovery time across all simulations per biome (black dots), as well as across a subset of simulations (coloured squares and triangles). The differences were first calculated for each grid-cell and only then averaged over biomes, thereby providing an estimate of the

relative importance of former LU duration versus former LU type on recovery times. The black dots show the sensitivity of recovery times to LU history across all simulations for each biome. The red, blue and green squares indicate the relative contribution of LU type to this sensitivity, and the orange and purple squares the relative contributions of pasture and of cropland duration. While substantial differences occur across the pasture simulations in tropical forests, savannas and grasslands and across croplands in boreal forests (thereby emphasising the importance of LU duration in these regions),

major differences between P100 and C100 occur in boreal forests and grasslands (thereby emphasising the importance of LU type if agricultural duration was long). On the other hand, in our simulations dominant PFT recovery in temperate forests is hardly influenced by the type of former LU, or, conversely, pasture duration has negligible effects on boreal forest recovery.

### 3.3    Vegetation C recovery

Compared to dominant PFT LAI, the starting point of vegetation C recovery averaged over all grid-cells is lower (11-14 %

of the reference simulation for vegetation C compared to 39-47 % for dominant PFT) due to higher percentage loss of vegetation C during the period of agriculture and recovery occurs slightly later (Fig. 2). Spatial patterns look more homogeneous than for dominant PFT (Fig. 3). While most grasslands recover within a few decades for all simulations, in particular so for post-cropland recovery, recovery occurs only after several decades or centuries in forest ecosystems. Lower standard deviations for the mean differences in recovery times for most biomes (Fig. 6) reflect the more uniform response of

post-agricultural vegetation C accumulation across different sites compared to dominant PFT recovery. Exceptions are tropical forests and grasslands, where the standard deviation is higher.

Significant differences in recovery times occur between simulations of different LU types that have the same duration, and between simulations of the same LU type but with different duration. For example, in the grasslands and savannas of southern, eastern and northern Africa, former croplands recover much faster than former pastures (see also Table 1 and Fig.

A2), because post-agricultural N availability is enhanced in these regions (Fig. 5). In former croplands in these environments, the combined effect of fertilizing and harvest is a net N flux to the ecosystem (not shown) and mineralization rates are enhanced after cropland abandonment (Fig. 4). We interpret this net N flux to the ecosystem as originating from high levels of water stress in these savannas and grasslands, resulting in greater C and N allocation to roots relative to leaves and thereby decreased harvest removal in this region (Fig. A3). Conversely, recovery in northern European forests is delayed

for C60 and, to an even greater extent, C100 because in this region N removal by annual harvest exceeds N addition through fertilization during the agricultural period (not shown) and post-agricultural N mineralization rates in this region are substantially reduced compared to the other simulations many decades or even a few centuries after abandonment (Fig. 4).



Differences in vegetation recovery times resulting from agricultural duration are mostly found in temperate and boreal forests for the cropland simulations (here longer durations result in longer recovery times due to reduced N availability, Fig. 5) and in tropical forests and shrublands for the pasture simulations, thereby emphasizing the importance of agricultural duration in these regions (see also Fig. 6).

5    ## 3.4    Soil C recovery

Relative depletion of soil C content under crop and pasture LU is not as large (loss of 0-11 % compared to the reference simulation) as for vegetation C (Fig. 2). However, regeneration proceeds over longer time scales due to slower C accumulation in soils than in vegetation. C depletion is generally more pronounced for former crops than for pastures due to the greater harvest efficiency, which leads to more biomass removed each year, and the effect of tillage enhancing soil 10    respiration (Sect. 2.1). Upon re-conversion, soil C accumulation is delayed for the pasture simulations, especially for P20 where the residual roots and other litter left after the original deforestation event continue to decay and soil C decreases for some decades. The general delay for pastures is associated with larger heterotrophic respiration rates (not shown) compared to rates calculated in recovering croplands and low litter input in the early stage of regrowth in forested ecosystems.

Soil C recovery rates are highly latitude-dependent (Fig. 3), being much slower in temperate (~250 years) and boreal forests 15    (~400 years) than in the tropics (<100 years, sometimes even within five years). Initial soil C depletions are larger in higher latitudes, whilst these regions also suffer from low productivity, thereby reducing C input to the soil upon regrowth. Additionally, in the intensive LU simulations (P100, C60, C100), vegetation productivity in the boreal region is further reduced compared to the reference simulation in the first 200 years of regrowth (not shown), reducing litter input to the soil even further.

20    Soil C recovery times differ substantially between simulations in many areas. LU type is particularly important in grasslands and non-tropical forests. While croplands tend to recover faster than pastures in grasslands of southern and northern Africa, the opposite occurs in most temperate and boreal forests, but also the northern Sahel, where soil C after re-conversion from croplands does not recover at all. Post-agricultural N availability is enhanced in parts of the Sahel for the cropland simulations due to increased N mineralization rates (Fig. 4 and Fig. 5), and trees benefit more than grasses, leading to a shift 25    in the equilibrium vegetation state towards woody species (not shown), which results in an overall lower soil C pool size. Counter to *a priori* expectations, for tropical and temperate forests and for shrublands, the difference between P20 and C20 is usually higher than between P60 and C60, or P100 and C100 (Fig. 6). Pasture duration is relevant for speed of soil C recovery in most ecosystems and, apart from in the tropics, a longer duration usually delays recovery, mainly due to substantial initial depletions after long pasture durations (Fig. 2). For croplands, longer durations tend to delay recovery in 30    temperate and boreal forests but accelerate soil C recovery in the (sub)tropics. This is somewhat unexpected for the tropical forest biome where longer cropland durations usually do not increase N availability upon abandonment in our simulations



(Fig. 5). However, while tropical soils lose large amounts of C during the first decades of cropland use, slow C accumulation takes place thereafter, resulting in higher soil C values at the end of the agricultural period for C100 than for C20 in large parts of eastern Africa. This is different to temperate and boreal forest where soil C decreases throughout the entire cropland period. Overall, the greatest sensitivity of soil C recovery times to different LU histories is found in boreal forests and grasslands where maximum differences across simulations are often several centuries (Fig. 6). While maximum differences in boreal forests are mainly due to differences across simulations of same LU type but different duration, the sensitivity of grasslands is mainly due to differences between simulations of different LU type, thereby emphasizing the importance of duration and type of agriculture in a range of biomes.

## 3.5 NBP recovery

NBP switches from being a C source to the atmosphere during the period of land management to a C sink after reconversion to natural vegetation (Fig. 2). The sink capacity of the re-growing vegetation is greatest during the first decades and then gradually goes back to the NBP of the reference simulation. P20 and, to a lesser extent, C20, act as a smaller sink than the other simulations. Recovery generally occurs slower in temperate and boreal regions than in the tropics for all simulations (Fig. 3). Apart from boreal forests, standard deviations of mean differences in recovery times are very small in all biomes compared to the other variables (Fig. 6). Recovery times are often somewhat lower than those which would be expected from vegetation and soil C recovery times. This is because the greater standard deviation of NBP in our reference simulation (Fig. 2) reduces the threshold value in our recovery definition, thereby making it easier to reach recovery levels for NBP. We discuss the implications of this further in section 4.2.

Differences in NBP recovery times between simulations are relatively small (typically some years to few decades, see Table 1). The largest differences in recovery times are found in the boreal forests between the cropland simulations, and, as for soil C, the differences are often greater between P20 and C20 than between P100 and C100 (Fig. 6).

## 4 Discussion

### 4.1 Comparison to observations and previous studies

The effects of forest conversion to croplands or pastures are relatively well studied. Tilled croplands typically show large depletions of soil C compared to natural forest vegetation, but the picture for pasture is more diverse (Davidson and Ackerman, 1993; Don et al., 2011; Guo and Gifford, 2002). A global meta-analysis of Guo and Gifford (2002) reported, averaged over different sampling depths, a 42 % loss of soil C when native forest was converted to cropland while soil C increased by 8 % when native forests were replaced by pastures. These patterns were qualitatively reproduced also by LPJ-GUESS, albeit in a version without coupled N dynamics (Pugh et al., 2015). However, no information for specific biomes or



time since conversion were provided in the Guo and Gifford study, in which most of the analyzed sites were located on the American or Australian continent. Their reported average soil C loss in croplands is higher than even in our C100 simulations (-17 % to -12 % for our forest biomes). Don et al. (2011) asserted average soil organic C losses of 25 % at a mean sampling depth of 36 cm 28 years after tropical old-growth forests had been converted to croplands. When conversion

was into pastures, soil C losses were 12 % at 36 cm depth after 25 years. These values are closer to our simulations which averaged a loss of around -11 % and -12 % in tropical forests for C20, respectively C60, and +2 %/-4 % for P20/P60. Pugh et al. (2015) studied the C dynamics of soils in managed lands in LPJ-GUESS and found C accumulation even after 100 years of grazed pasture at some locations, especially for low atmospheric $CO_2$ concentrations. However, they used the C-only version of the model, thereby neglecting C-N interactions and increased N limitation on grass growth with time due to

N removal by harvest. For croplands, in their study explicitly represented by a number of managed but unfertilized crop functional types, soil C reductions in Europe and Africa were much greater after 100 years (~50 %) than in our case (~12 %), possibly partly because tillage has different effects in the two soil models applied.

By contrast to studies of LU effects compared to previously natural ecosystems, the regeneration of ecosystems after agricultural abandonment has been studied less, and a direct comparison to our simulations is challenging, either because

limited information about former LU or reference conditions was provided in these studies, or because there are important differences from our setup in terms of management and LU duration. Additionally, most of the available studies were conducted in Amazonia or North America (Don et al., 2011). Many studies focus on the recovery of biodiversity or species richness (Cramer et al., 2008; Queiroz et al., 2014), but these variables cannot adequately be captured by our large-scale PFT approach. It is often assumed that the ecosystem will gradually return to its previous state and that intensive LU delays

recovery but the time scales are widely unknown and differ across variables and regions, e.g. tropical species composition recovers much slower than forest structure and soil nutrients (Chazdon, 2003). Different recovery processes are strongly interlinked, e.g. vegetation accumulation and turnover are key factors in the replenishment of soil quality and nutrients which in turn determine plant productivity, and post-agricultural soil C and N dynamics have been shown to correlate during the regeneration of ecosystems (Knops and Tilman, 2000; Li et al., 2012).

Saldarriaga et al. (1988) estimated a necessary time period of 190 years for previously slash-and-burned agricultural sites to reach above-ground biomass characteristic of mature tropical forest (compared to 121 ± 65 years in our C20 simulation) but such estimates are highly uncertain due to the large variability of physical and biotic characteristics as well as of land management in tropical forests (Kauffman et al., 2009). The sequestration rate of above-ground biomass in regenerating tropical forests was reported to slow down some decades after agricultural cessation (Silver et al., 2000), while linear C

accumulation was found on sites in the US (Hooker and Compton, 2003) and England (Poulton et al., 2003) during the first 115-120 years of temperate forest regrowth, even though in the latter case biomass started to evolve only after around two decades. LPJ-GUESS shows a reduction of vegetation C accumulation with time for both biomes, even though less pronounced than in the Silver (2000) study. Uhl et al. (1988) analyzed the influence of pasture history on secondary forest



regeneration in eastern Amazonia. They found lower biomass accumulation rates and reduced species richness after heavy use compared to light use. While light use implied low grazing pressure and no weeding, clearing in heavy use pastures involved several cutting and burning episodes and bulldozing, followed by moderate grazing. Pasture intensities also differed in terms of LU duration (0-13 years). Hughes et al. (1999) confirmed the inverse relationship between LU duration and

biomass regeneration in tropical secondary forests for agriculture in general (mostly fields but sometimes alternating between crops and pasture): While sites that experienced a short period of LU (1-7 years) reached primary forest levels with about 90 % of their above-ground biomass after only 31 years, recovery was delayed to 79 years for prolonged periods of LU (13-30 years). Delayed recovery of woody vegetation (in terms of basal area and maximum tree height) was also reported for longer cropland durations in eastern Madagascar (Randriamalala et al., 2012).

Our study indicates that tropical forest biomass indeed recovers slower after longer periods of agricultural use, even though the effect is relatively small when the previous LU type is cropland (average recovery time of 121 years for C20 and 139 years for C100). Regarding effects of LU type, it is known that tropical vegetation recovers slower for former pastures than for swidden agriculture (Chazdon, 2014; Moran et al., 2000; Silver et al., 2000) even though early studies often did not adequately account for differences in soil texture and climate across sites (Zarin et al., 2001). Wandelli and Fearnside (2015)

reported a 38 % slower aboveground biomass accumulation for former pastures than for slash-and-burned croplands in Amazonia after up to 10 years of agriculture. However, while in our simulations tropical vegetation recovery was faster for former croplands than for former pastures after long agricultural periods (P100 and C100), we found a slower recovery in croplands than in pastures after short durations (P20 and C20), apparently resulting from nitrogen dynamics. Overall, the available studies that looked at vegetation recovery upon abandonment indicate that biomass accumulation slows down after

some decades and that accumulation rates correlate negatively with agricultural duration. Our simulations show that the rate of vegetation C sequestration indeed declines over time and that longer LU durations delay recovery in each of the analyzed biomes. Observations also indicate that use of land for pasture delays recovery in the tropics upon pasture abandonment, compared to cropping, but in our simulations this seems to be the case only after long agricultural durations.

Several reviews studied soil C dynamics after agricultural abandonment, however, interpretation is often hindered by

combining different soil layers or aggregating different LU types (Li et al., 2012), and by large variations observed across studies (Post and Kwon, 2000). A global review of 43 studies (Paul et al., 2002) reported high variability in soil C changes after afforestation of abandoned agricultural systems with the tendency to lose C in the surface soil during the first years of regrowth, followed by a decrease in the rate of decline and finally exceeding former agricultural levels after around 30 years. Greatest post-agricultural C losses occurred in former pastures in the temperate zone while former crops, especially in the

tropics, usually accumulated C already in the first years upon reconversion. Laganiere et al. (2010) asserted that the positive effect of afforestation on soil organic C was more pronounced for former croplands than for pastures and that boreal ecosystems require a longer time period to accumulate C. Poeplau et al. (2011) reported fast soil C depletion and a new equilibrium level 23 years after temperate forests were converted to cropland, followed by slow soil C accumulation after



conversion back to forests, with no new equilibrium reached after 120 years. All of these patterns are reproduced in our simulations, suggesting that LPJ-GUESS captures the salient processes, with the exceptions that soil C decreases throughout the entire cropland duration in temperate forests and that it can take several centuries to reach the soil C content of former pastures again.

Assuming constant C accumulation, time periods necessary to reach pre-cropland levels after abandonment have been estimated to be 158 years in the surface 60 cm for a native prairie (Potter et al., 1999), 230 years at 0-60 cm depth for a temperate oak savanna (Knops and Tilman, 2000), at least 100 years at 0-10 cm depth for a mixed temperate forest (Foote and Grogan, 2010), and 50-60 years at 0-25 cm depth for tropical forests (Silver et al., 2000). However, detailed information about cropland duration were rarely provided. Schierhorn et al. (2013) used the LPGmL DGVM to study short-term C
dynamics caused by cropland abandonment in the former Soviet Union. They found a small C source during the first years of regrowth caused by low productivity and soil emissions, followed by accelerating soil C uptake a few years thereafter. In contrast, our results suggest immediate C sequestration in post-cropland soils. Altogether, even though discrepancies exist across studies, observations indicate greater initial soil C depletion for croplands, but also higher accumulation rates after abandonment. Both of these patterns are reproduced by our simulations. The impact of LU duration was rarely studied,
however, our results suggest that even though longer agricultural durations mostly result in greater initial soil C depletions, recovery can occur at similar or even faster speed in the (sub)tropics. In temperate and boreal forests long LU durations tend to delay recovery.

The LPJ-GUESS model has been successfully tested against a range of observations and observation-based products, including vegetation distribution and dynamics, and soil C response to changes in vegetation cover (Hickler et al., 2004;
Miller et al., 2008; Pugh et al., 2015; Smith et al., 2014). In our simulations, we used only two different agricultural land cover types (intensive grazing and fertilized, tilled crops). Our analysis would therefore not identify effects of, for instance, clearing technique (e.g. burning *cf.* mechanical removal) or different land management practices (e.g. repeated burning or irrigation) within one land cover type. For example, recovery of species richness and maximum tree height of secondary forests occurs faster under no tillage compared to heavy tillage (Randriamalala et al., 2012).

Our study is intended as a stylized experiment to highlight the importance of LU history on ecosystem state and fluxes across biomes. Still, some processes are not currently included in LPJ-GUESS with the potential to affect post-agricultural ecosystem recovery at least regionally. One aspect is the phosphorus cycle which is not implemented in LPJ-GUESS, even though it can be significantly altered by LUC (MacDonald et al., 2012; McLauchlan, 2006). And while C and N cycles interact in LPJ-GUESS (Smith et al., 2014), the uniform annual fertilizer rate we applied in this study might be realistic in
some regions, such as parts of Europe, but exceeds present-day fertilizer use in Africa (Potter et al., 2010). Seed availability, remnant trees and resprouting from surviving roots are important factors during initial stages of tree colonization following agricultural cessation (Bellemare et al., 2002; Cramer et al., 2008). While LPJ-GUESS does not account for these effects



explicitly, seedling establishment is limited by a suitable growth environment, such that effects like re-sprouting or remnant trees as seed sources are mimicked. The model has been shown to e.g., reproduce vegetation recolonization in northern Europe during the Holocene well (Miller et al., 2008) as well as canopy structural changes as a function of forest age (Smith et al., 2014). What is more, by using a prescribed climate in our simulations, hydrological biosphere-atmosphere interactions and feedbacks are not captured (Eltahir and Bras, 1996; Giambelluca, 2002), which could alter regional climate in response to land cover change, potentially affecting recovery rates, especially in tropical regions. Biophysical effects are not restricted to modifications of the water cycle but also include changes in surface albedo and roughness length as a function of ecosystem structure and composition, thereby affecting air mixing and heat transfer. While forests generally absorb more sunlight than grasslands (e.g. Culf et al., 1995), differences amongst tree species and age classes exist as well. Biophysical effects related to realistic land-use change tend to have limited effects on the global mean energy balance, but have been shown to have large local to regional impacts (Alkama and Cescatti, 2016; Peng et al., 2014), with the typical direction of the temperature change varying between boreal, temperate and tropical climate (Pielke et al., 2011). Additionally, while we focus in our analysis on C sequestration rates, there might be biogeochemical implications beyond C. For instance, the emissions of biogenic volatile organic compounds (BVOCs) to the atmosphere vary greatly amongst plant species (Kesselmeier and Staudt, 1999). BVOCs affect atmospheric composition and climate via ozone production, lengthening the lifetime of atmospheric methane, and contributing to secondary organic aerosol formation (Penuelas and Staudt, 2010; Wu et al., 2012). BVOC emission factors might also be drastically influenced by wildfires (Ciccioli et al., 2014), which in turn are driven by species composition and vegetation density. Thus, different successional trajectories of ecosystem structure and composition recovery have the potential to directly modify air quality and climatic conditions under which regrowth occurs, potentially creating positive or negative climate system feedbacks.

### 4.2    Implications of recovery definition

The term recovery is very subjective and, in the absence of a universal definition amongst ecologists, different approaches imply the potential to significantly modify absolute recovery times. By our definition we examine recovery from a C sequestration perspective and therefore do not capture situations e.g. in cases where the system approaches towards a new equilibrium (as soil C did in some regions in the cropland simulations). In order to obtain a better understanding of the uncertainties related to our definition we therefore explored four alternative plausible recovery definitions.

When applying a mean minus 2 σ threshold (instead of a mean minus 1 σ threshold), recovery times are generally shorter, e.g. on average 75 instead of 106 years for vegetation C in P60, but the overall geographic patterns are very consistent across both definitions (not shown). For all variables and simulations, notable differences between both definitions occur in regions with longest recovery times, especially for subtropical soil C in the pasture simulations.





Recovery based on percentage change (Fig. A4) results in more heterogeneous patterns across variables when compared to our standard recovery definition. Applying a threshold of 95 % of the mean, instead of a mean minus 1 σ threshold, produces slightly longer dominant PFT recovery times in parts of the temperate and tropical forests, and shorter recovery times in grasslands, especially for the pasture simulations. Vegetation C shows similar patterns to the dominant PFT, however, the differences are more pronounced. Soil C recovery times generally decrease dramatically, especially outside the tropics. NBP recovery times generally increase, particularly in forest ecosystems.

By checking not only if the value of the variable is above the reference mean minus 1 σ but also if the value is below the reference mean plus 1 σ, and with the "minimum rule" also applied to the maximum (Fig. A5), one can test which ecosystems recover from above rather than below background levels. Mostly grasslands are affected. Dominant PFT recovery under this definition takes slightly longer throughout the African grasslands for the pasture simulations, and considerably longer in parts of northern and southern Africa for the cropland simulations. Patterns are similar for vegetation C but more pronounced, especially for croplands. Soil C recovery is notably longer in subtropical and eastern African grasslands. NBP differences look similar to soil C but the effects are much smaller. We do not use an upper limit in our primary definition however, because, in the case of C storage, the ecosystem is already operating at a level of service above that which the unmodified ecosystem would have provided.

Finally, when using the mean ± 1 σ definition and additionally checking if the variable is still in the mean ± 1 σ range at the end of the simulation period (not shown), many grid-cells did not recover even within the set maximum cut-off of 800 years. Elements of random fluctuations due to natural variability made a clear identification of recovery period difficult in that case. In particular for soil C, no recovery is found for parts of eastern and subtropical Africa. The system converges towards a new equilibrium state in these regions which lies above reference values. NBP stays within background levels everywhere.

Altogether, the alternative recovery definitions agree on the general findings when applying our standard definition, especially in terms of relative recovery rates. For all definitions, vegetation C and LAI recover faster in grasslands than in forest-dominated ecosystems and soil C recovery takes much longer in higher latitudes. However, some areas, especially in the subtropics, "recover" from values higher than in the reference simulation and these cases are not captured by our standard definition. Additionally, in the tropics, soil C accumulation sometimes does not stop once background values are reached and soil C leaves the reference range. When recovery is defined based on standard deviation, NBP recovery is often quicker than recovery of the C pools. This inconsistency emphasizes the importance of both recovery definition and selected variables when studying the recovery of ecosystems (Jones and Schmitz, 2009). This is particularly relevant for flux tower measurements where ongoing regeneration might be overlooked due to large variability of NEE.

**5      Conclusions**



Most studies which explored the effects of distant human activities on present-day ecosystems were restricted by sampling difficulties, small spatial scales, short time periods since abandonment, and little information about background conditions or the specific LU history of the site. Here, we use a model-based approach to study the legacy effects of agricultural LU history (type and duration) on ecosystem regeneration and C sink capacity after the cessation of agriculture in a range of biomes across Europe and Africa. The model reproduces qualitatively the response found at study locations, including distinct differences in recovery between different variables of the terrestrial carbon cycle. Long-lasting legacy effects of former agricultural intensity emerge as important for present-day ecosystem functions. These findings have implications for various scientific applications:

1. Long-term monitoring sites (e.g. FLUXNET) and Earth observation systems need to collect and maintain detailed information about past and present land cover and land management to adequately interpret their data.

2. Assessments of trends in data from sites that seek to identify impacts of climate change and/or atmospheric $CO_2$ concentration need to make sure that legacy effects of past LU are not confounding the observed trends.

3. Simulation experiments need to move beyond deforestation but also represent in a more detailed manner re-growth dynamics following agricultural abandonment at the sub-grid level. At the moment a few DGVMs have started to do so (Shevliakova et al., 2009; Stocker et al., 2014; Wilkenskjeld et al., 2014) based on model products of tropical shifting cultivation (Hurtt et al., 2011), but accounting for gross land cover changes is also important in other regions like Europe (Fuchs et al., 2015). Failure to consider land-use history may lead to errors in the simulation of vegetation properties, potentially resulting in biases in carbon sequestration or energy balance calculations, with subsequent implications for simulations of regional and global climate.

4. Assessing the efficiency of climate mitigation through large-scale reforestation or afforestation projects will require knowledge about the type and duration of previous LU. Our simulations suggest that the potential to rapidly sequester C in biomass and soil is greatest in tropical forests following short periods of cropland while boreal forests accumulate C slowest, especially when previously used for pasture. Special attention should be given to monitoring changes in below-ground C, as in most places the accumulation of soil C is much more sensitive to LU history than C accumulation in re-growing trees.

5. Besides determining changes in C dynamics, pathways of secondary forest regrowth affect other nutrient cycles and atmospheric composition for instance through variable emissions of compounds like BVOCs. Furthermore, the net climate effect of reforestation and afforestation will not be restricted to geochemical effects but will also involve changes of land surface parameters, with effects on regional energy and water fluxes that can amplify or counteract climate benefits from C sequestration (Anderson et al., 2011).

**Appendix A**



**Acknowledgements**

This work was funded by the Helmholtz Association through the International Research Group CLUCIE and by the European Commission's 7[th] Framework Programme, under Grant Agreement number 603542 (LUC4C). This work was

supported, in part, by the German Federal Ministry of Education and Research (BMBF), through the Helmholtz Association and its research program ATMO.

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

Table 1: Average recovery times and standard deviations per biome and for each simulation. Recovery times are depicted in Figure 4.

| Biome | Simulation | | | | | |
|---|---|---|---|---|---|---|
| | P20 | P60 | P100 | C20 | C60 | C100 |
| | **dominant PFT** recovery time, averaged per biome | | | | | |
| Tropical Forest | 90 ± 55 | 112 ± 48 | 121 ± 50 | 113 ± 54 | 125 ± 52 | 126 ± 51 |
| Temperate Forest | 102 ± 74 | 96 ± 63 | 93 ± 57 | 99 ± 71 | 89 ± 61 | 92 ± 69 |
| Boreal Forest | 47 ± 89 | 52 ± 97 | 53 ± 90 | 47 ± 95 | 60 ± 111 | 145 ± 178 |

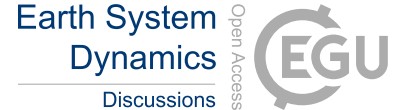

| Savanna | 47 ± 71 | 57 ± 74 | 62 ± 77 | 50 ± 65 | 57 ± 73 | 59 ± 76 |
|---|---|---|---|---|---|---|
| Shrub | 95 ± 93 | 104 ± 101 | 108 ± 100 | 103 ± 100 | 109 ± 112 | 109 ± 112 |
| Grassland | 76 ± 108 | 102 ± 109 | 115 ± 109 | 45 ± 77 | 55 ± 97 | 58 ± 100 |
| Total | 80 ± 85 | 93 ± 84 | 99 ± 84 | 77 ± 78 | 83 ± 85 | 90 ± 95 |
| | **Vegetation C** recovery time, averaged per biome | | | | | |
| Tropical Forest | 106 ± 50 | 137 ± 61 | 150 ± 65 | 121 ± 65 | 138 ± 73 | 139 ± 74 |
| Temperate Forest | 84 ± 24 | 93 ± 31 | 108 ± 46 | 91 ± 29 | 124 ± 59 | 149 ± 79 |
| Boreal Forest | 102 ± 47 | 113 ± 57 | 127 ± 71 | 111 ± 55 | 144 ± 79 | 187 ± 107 |
| Savanna | 49 ± 37 | 61 ± 44 | 66 ± 46 | 35 ± 40 | 42 ± 43 | 43 ± 44 |
| Shrub | 73 ± 40 | 86 ± 48 | 96 ± 51 | 60 ± 38 | 69 ± 48 | 73 ± 54 |
| Grassland | 96 ± 136 | 119 ± 140 | 126 ± 138 | 40 ± 98 | 43 ± 102 | 45 ± 105 |
| Total | 88 ± 80 | 106 ± 87 | 117 ± 90 | 75 ± 74 | 92 ± 87 | 101 ± 98 |
| | **Soil C** recovery time, averaged per biome | | | | | |
| Tropical Forest | 74 ± 60 | 69 ± 43 | 66 ± 45 | 80 ± 46 | 64 ± 46 | 49 ± 43 |
| Temperate Forest | 207 ± 98 | 229 ± 105 | 241 ± 117 | 237 ± 108 | 261 ± 133 | 260 ± 144 |
| Boreal Forest | 327 ± 107 | 381 ± 122 | 421 ± 140 | 362 ± 112 | 425 ± 132 | 454 ± 161 |
| Savanna | 84 ± 132 | 132 ± 191 | 162 ± 233 | 85 ± 112 | 83 ± 125 | 74 ± 126 |
| Shrub | 107 ± 140 | 129 ± 161 | 135 ± 168 | 137 ± 173 | 139 ± 183 | 125 ± 183 |
| Grassland | 286 ± 234 | 366 ± 262 | 422 ± 283 | 239 ± 227 | 219 ± 229 | 198 ± 228 |
| Total | 182 ± 176 | 220 ± 209 | 245 ± 236 | 182 ± 171 | 183 ± 186 | 174 ± 194 |
| | **NBP** recovery time, averaged per biome | | | | | |
| Tropical Forest | 57 ± 37 | 65 ± 26 | 71 ± 27 | 56 ± 28 | 64 ± 24 | 65 ± 24 |
| Temperate Forest | 97 ± 29 | 108 ± 29 | 113 ± 31 | 102 ± 30 | 112 ± 31 | 119 ± 36 |
| Boreal Forest | 136 ± 55 | 146 ± 56 | 152 ± 58 | 139 ± 54 | 151 ± 59 | 169 ± 71 |
| Savanna | 31 ± 40 | 34 ± 30 | 36 ± 26 | 29 ± 18 | 32 ± 17 | 33 ± 17 |
| Shrub | 51 ± 37 | 58 ± 31 | 59 ± 29 | 52 ± 27 | 58 ± 26 | 59 ± 25 |
| Grassland | 25 ± 37 | 31 ± 31 | 35 ± 30 | 27 ± 15 | 34 ± 20 | 36 ± 22 |





| Total | 59 ± 51 | 66 ± 49 | 71 ± 49 | 60 ± 45 | 68 ± 47 | 72 ± 52 |
|---|---|---|---|---|---|---|




Figure 1: Vegetation C [kg C m$^{-2}$] for the reference simulation, averaged over the whole simulation period of 900 years (upper left), soil C [kg C m$^{-2}$] (lower left), dominating PFT (upper right), and corresponding biomes (lower right). Gridpoints with a NPP below 0.1 kg C m$^{-2}$ yr$^{-1}$, deserts and tundra, and latitudes above 62.5°N are masked in grey.

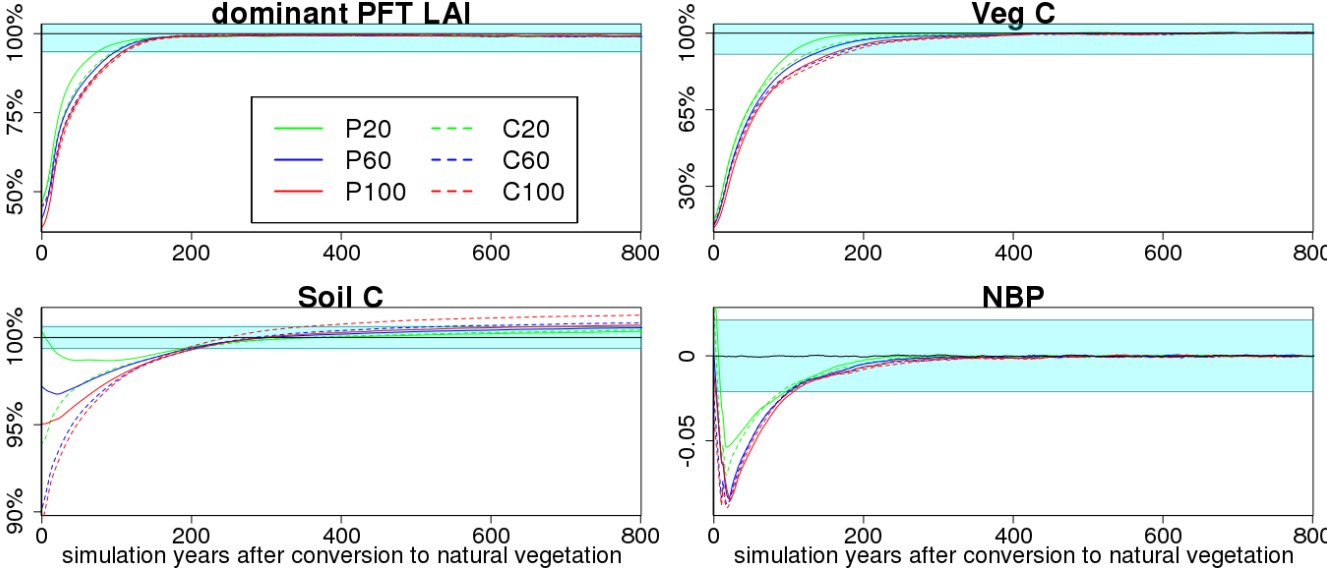

Figure 2: Time series (20 year running mean) of dominant PFT LAI, vegetation C, soil C and NBP for the different experiments, starting from the time of reconversion to natural vegetation and area-averaged over all grid-cells. PFT, vegetation C and soil C are shown in relative values compared to reference simulation mean while NBP is shown as absolute values [kg C m$^{-2}$ yr$^{-1}$] because values cannot be presented relative to a zero background. The cyan-shaded area corresponds to reference simulation mean ± 1 σ. Note the different scales on the y-axes.





Figure 3: Maps of recovery times in years for dominant PFT, vegetation C, soil C, and NBP, exemplarily for the P20, P100, C20, and C100 simulation.





Figure 4: Average net N mineralization rates [kg N ha$^{-1}$ yr$^{-1}$] in the soil during the first 100 years of regrowth, exemplarily for the P20, P100, C20, and C100 simulation.



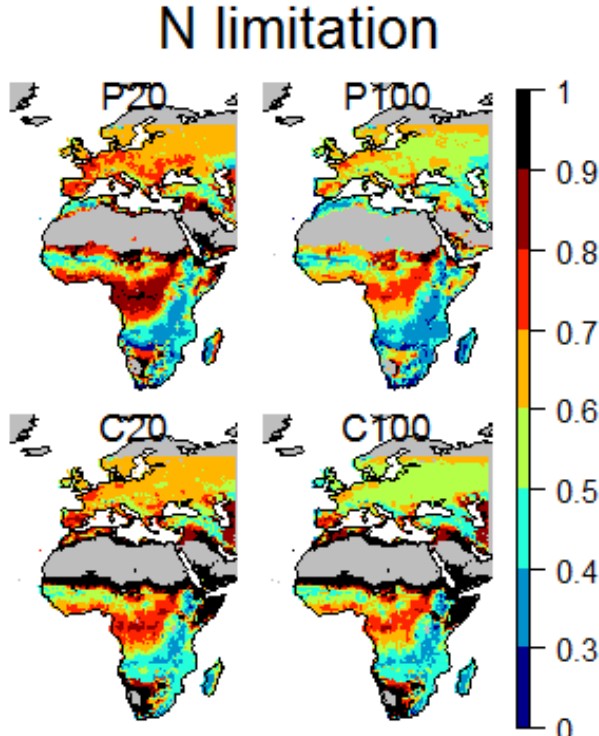

Figure 5: Average N limitation on vegetation rubisco capacity (and thus on GPP) during the first 100 years of regrowth, exemplarily for the P20, P100, C20, and C100 simulation. N limitation is a number scaling from 0 (completely N limited) to 1 (no N limitation) (Smith et al., 2014).





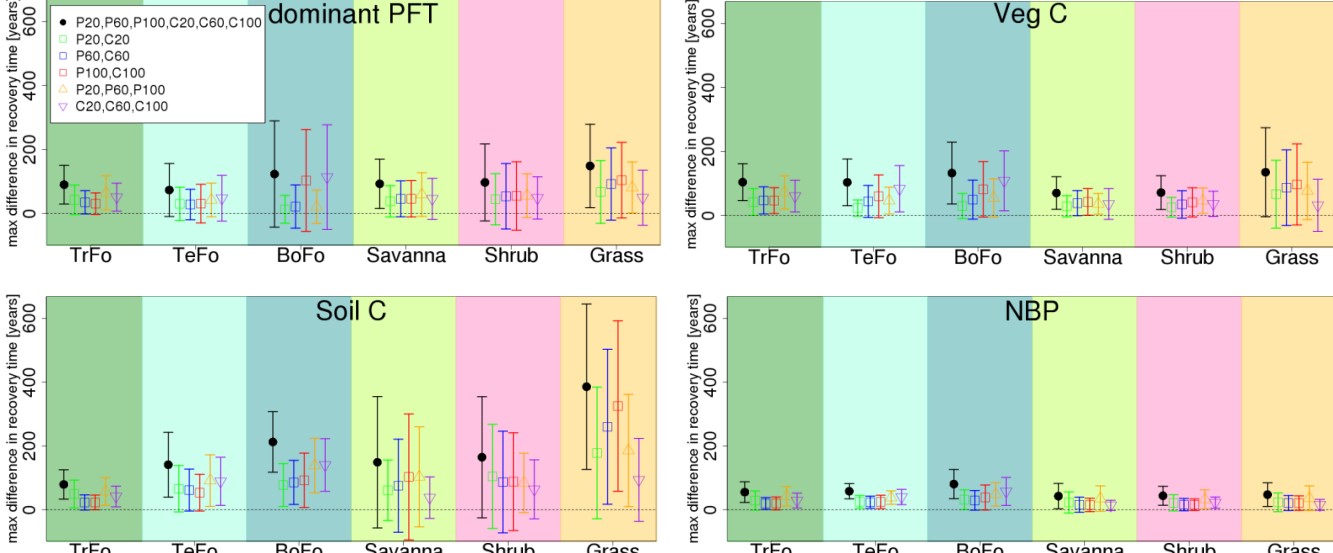

Figure 6: Maximum difference in recovery time (longest recovery time minus shortest recovery time of all selected simulations) for dominant PFT, vegetation C, soil C, and NBP. Black dots show maximum differences across all six simulations (P20, P60, P100, C20, C60, C100), green squares differences across 20yr pasture and cropland simulations (P20, C20), blue squares differences across 60yr pasture and cropland simulations (P60, C60), red squares differences across 100yr pasture and cropland simulations (P100, C100), orange triangles differences across pasture simulations (P20, P60, P100), purple triangles differences across cropland simulations (C20, C60, C100). Background colors indicate associated biomes, arrows one standard deviation, the dashed line 0 years difference.





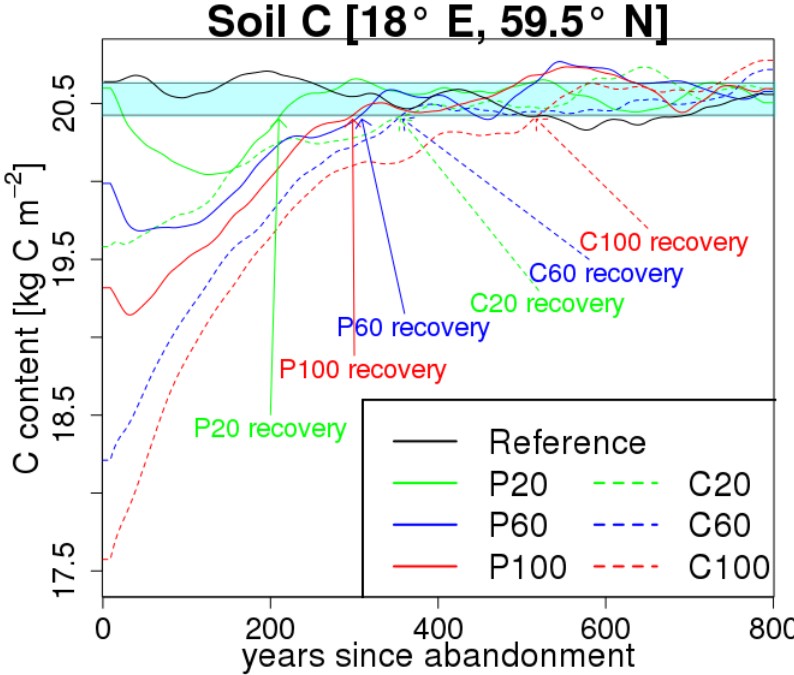

Figure A1: Soil C for the six simulations after conversion to natural vegetation at one single example site to illustrate how recovery time was calculated according to our definition. The cyan-shaded area corresponds to reference simulation mean ± 1 σ. When soil C exceeds the mean - 1 σ threshold and the time of the minimum (which in this case is located in the first 200 years and below the mean - 1 σ threshold for all six simulations) is passed, recovery is achieved.











Figure A2: Histogram of recovery times for dominant PFT, vegetation C, soil C, and NBP for the six experiments. Colors indicate different biomes.




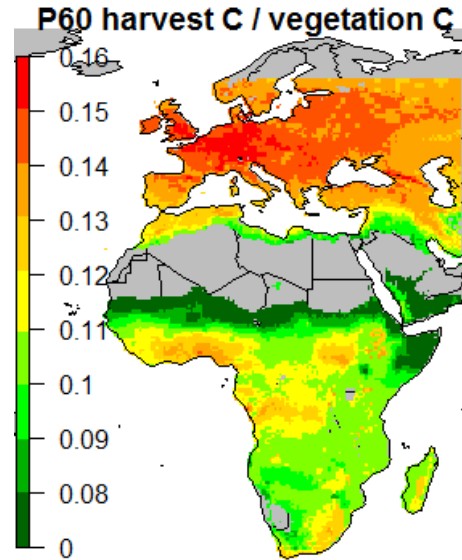

Figure A3: Annual ratio of C removed by harvest and C stored in vegetation, averaged over the whole agricultural period and exemplarily for P60. As only above-ground biomass is harvested, lower values indicate increased C allocation to roots compared to leaves due to limited water supply.



Figure A4: Maps of recovery time for dominant PFT, vegetation C, soil C and NBP with an alternative recovery definition, exemplarily for the P60 and C60 simulation. The definition is the same as our standard definition but with a mean * 0.95 threshold instead of mean - 1 σ.




Figure A5: Maps of recovery time for dominant PFT, vegetation C, soil C and NBP with an alternative recovery definition, exemplarily for the P60 and C60 simulation. The definition is the same as our standard definition but with a mean ± 1 σ threshold and the minimum check also applied to the maximum instead of a mean - 1 σ threshold and only checking the minimum.