# Peer review of "Impacts of land-use history on the recovery of ecosystems after agricultural abandonment"

_Earth System Dynamics, 2016_

## Referee Comment (RC1) · Anonymous Referee #1 · 9 May 2016

This paper provides an interesting model assessment of the time taken for land to 'recover' after a conversion from agricultural back to 'natural' land. This is a solid, if unsurprising, paper. The results seem robust and the background well researched. Some of the conclusions over-reach a little. It is well worthy of publication. There are a few minor improvements I would recommend.

The introduction is rather long for a paper with this amount of results, and many of the points are repeated in the discussion. Ideally, each point should only be made once, and some rationalization of the amount of general background given would be good too.

p 2, line 7. Who is the author of this editorial? The original source/s would be better.

p 4, line 14 and 20. The amount of N fertilization to crops and how much is 'left behind'

seems to be a critical aspect of the story, as N limitation becomes important in the subsequent LUC back to 'natural' (e.g. the discussion on p 7/8, lines 22 - 4). Some context of the size of this (where did the fertilization value come from?), and discussion of this assumption's effects on the results are necessary.

p 5, line 10. I'm a little unsure about the exclusion of desert and tundra. If all grid-cells above 62.5N are excluded from the results, why are they included in the simulations, or the results plots? Why not just have the map end at 62.5N and save the space?

With regards to the desert, I'm wondering what (if any?) representation of irrigation there is, and whether if there is irrigation, 'desert greening' with irrigated cropland might be an interesting aspect of this study.

p 6, line 19. Check the sense of this sentence.

p 7, lines 5 - 8. This ought to be in the figure caption, not the main text.

p 9, lines 1-3. This tropical soil carbon response is interesting - can you enlarge on what the physical or model mechanism that causes it is? What causes the change from soil carbon loss to accumulation?

The discussion is very long, rather dry, and as a reader it is difficult to get a clear overall sense of how well the model results compare to field observations. The simple fix for this would be a table with: observation type (e.g. soil carbon recovery time in pasture); observation value (e.g. 100 years); closest model (e.g. P20); closest model value (e.g. 50 years); model performance (either + (too high), - (too low), or a tick (within the obs. uncertainty)). This would convert about 4 pages of hard-to-digest discussion into 1 page of at-a-glance clear results.

Please could you define acronyms and unusual terms, when they are first used? e.g. swidden, NEE, NBP.

p 13, line 10. I'm not sure it's accurate to dismiss LUC biogeophysics as irrelevant at a global scale. (For where biogeopysics has a significant impact on the global climate,

**ESDD**
see for instance, Davies-Barnard et al., 2014; Davin and de Noblet-Ducoudré, 2010; Jones et al., 2012; Matthews et al., 2004; Pongratz et al., 2010.)

p 13, line 22. III defined is more the case than subjective.

p 13, line 24. Check the sense of this sentence.

p 14, lines 7 - 15. Why is  $\pm$  1 sd not the default way of analysis in this study? Wouldn't that make much more sense?

The conclusions need to be more specific, and restricted to results that can be directly evidenced from the results in the paper.

For instance, in #5, BVOCs are the main point being made - but BVOCs aren't included in the model or the paper, and the point is referenced elsewhere. If you have to reference another paper in your take home messages, you really should rethink what \*your\* take home messages are, because currently, they are someone else's.

Conclusion #3 also particularly suffers from lack of evidence. The results here show that for most variables, an equilibrium simulation of 100 years is plenty to sort out any legacy LUC effects. Soil carbon is an exception, but then there is lots of evidence that soil carbon is very uncertain in both models and observations. Longer simulations would do the same as re-growth dynamics in many cases, and so you need to highlight which variables in which regions under what time-scales, you have shown are affected.

The color schemes, especially on the maps, are not all that easy on the eye, and would be very difficult for someone who is color-blind to interpret. Could you consider another color scheme? You could look to http://colorbrewer2.org/ for some good, easy color schemes.

References:

Davies-Barnard, T., Valdes, P. J., Singarayer, J. S. and Jones, C. D.: Climatic impacts of land-use change due to crop yield increases and a universal carbon tax from a scenario
model, J. Clim., 27(4), 1413–1424, doi:10.1175/JCLI-D-13-00154.1, 2014.

Davin, E. L. and de Noblet-Ducoudré, N.: Climatic Impact of Global-Scale Deforestation: Radiative versus Nonradiative Processes, J. Clim., 23(1), 97–112, doi:10.1175/2009JCLI3102.1, 2010.

Jones, A. D., Collins, W. D., Edmonds, J., Torn, M. S., Janetos, A., Calvin, K. V., Thomson, A., Chini, L. P., Mao, J., Shi, X., Thornton, P., Hurtt, G. C. and Wise, M.: Greenhouse Gas Policy Influences Climate via Direct Effects of Land-Use Change, J. Clim., 26(11), 3657–3670, doi:10.1175/JCLI-D-12-00377.1, 2012.

Matthews, H. D., Weaver, A. J., Meissner, K. J., Gillett, N. P. and Eby, M.: Natural and anthropogenic climate change: incorporating historical land cover change, vegetation dynamics and the global carbon cycle, Clim. Dyn., 22(5), 461–479, doi:10.1007/s00382-004-0392-2, 2004.

Pongratz, J., Reick, C. H., Raddatz, T. and Claussen, M.: Biogeophysical versus biogeochemical climate response to historical anthropogenic land cover change, Geophys. Res. Lett., 37(8), L08702, doi:10.1029/2010GL043010, 2010. **ESDD**

---

## Referee Comment (RC2) · Anonymous Referee #2 · 24 May 2016

This manuscript evaluates ecosystem recovery after disturbance from agriculture and pasture management using a dynamic vegetation model and idealized case studies. This is an interesting application of the LPJ-GUESS model, however the results are not particularly novel. The study was well-executed and the paper is clearly written. I suggest publication with some minor revisions mainly to improve clarity.

General comment: I find the inconsistency in terminology relative to "vegetation composition" to be somewhat confusing. First the term vegetation composition is used (abstract); later it is defined as the LAI of the dominant PFT (page 3, L 15), and then finally in section 2.4 the reader learns that it is both the dominant PFT and the dominant PFT LAI. Therefore, it's not clear which result is presented in the figures since it seems to flip from LAI in figure 2 and dominant PFT in subsequent figures. It should be obvious to the reader how dominant PFT is defined and kept consistent throughout

the manuscript.

Page 5, L 29-30: reads awkwardly, to what does lower limit refer? Assuming that 'lower limit' implies the time to return to pre-disturbance conditions, wouldn't it be more appropriate to say that the ecosystem might never return. It is possible those sites have reached new equilibriums; does a trend still exist or has the model reached a new steady state?

Page 7, L1-2: This is interesting, looking at Figure 3 – there is a line of grids in the boreal forest that has exceptionally long time to recover LAI right next to grid cells that only take a fraction of that time to recover. The authors mention this, but provide no reason for the behavior. Looking at figure 5, both P100 and C100 have similar levels of N limitation in this region, so why does C100 take so much longer than P100 to recover?

Page 7, L14-16: This sentence should be rewritten for clarity.

Page 7, L17-18: The fact that grasslands return so quickly isn't surprising given the model setup. Although the authors don't specify how crops are modeled (other than some C and N modifications), I suspect that they are otherwise modeled exactly the same as grasses. If this is the case, grassland recovery would be almost immediate relative to dominant PFT and vegetation carbon.

Page 7, L21: Boreal forests seem to have a higher standard deviation than tropical in figure 6.

Page 8, L 6: The soil C loss of 0-11% seems very low, especially for agriculture lands. I would expect a minimum of 20% loss (even for a short 20 year period). This seems to be confirmed in the discussion section.

Page 8-9, L29-31-L1-3: This is unexpected and deserves some explanation. I'm also curious, how is soil C recovering so quickly for P100 and C100 (for example in central Africa) when the other components (LAI, Veg C, NBP) are taking much longer (a

ESDD
hundred years or more).

Page 10-13, Section 4.1: This section tends to ramble and focus on elements that are not related to this study. At times, it is difficult to read and should be shortened and focused.

Page 10, P1: The authors did a poor job of comparing the model output with observations. First, the authors never mention the soil depth of the model, so I don't think any of the comparisons with observations in this section are useful. Second, comparing against other versions of the model that didn't have the N cycle doesn't add any useful information regarding model performance capturing soil C loss after disturbance. I understand that soil C loss isn't the focus of the paper, but if the reader can't trust the soil c loss from management (which I suspect is underestimated), how can they trust the recovery estimates.

Page 10, L 28-32: Reword for clarity.

Page 11, L 29-30: This sentence is not clear.

Conclusion: I think one conclusion that wasn't made (that could be based on the alternate recovery results) is that recovery for some variables doesn't seem to be ever reached, only a new equilibrium (particularly for soil carbon).

Table 1: caption reads "Recovery times are depicted in Figure 4" – should be Figure 3.

Table 1: For the dominant PFT LAI recovery, I find it interesting that for a temperate forest, the P100 and C100 take less time to recover than the P20 and C20. Although all times are within the error bars, it still is not consistent with the other biomes. The authors don't mention this behavior in Section 3.2, but it would be nice to have an explanation.

Figure 1 has a lot of acronyms that aren't defined, please define each PFT.

---

## Author Comment (AC1) · 23 Jun 2016

We thank the two reviewers for their comments which certainly helped to improve the structure and readability of our manuscript. In the following we answer their comments.

Reviewer #1:

This paper provides an interesting model assessment of the time taken for land to 'recover' after a conversion from agricultural back to 'natural' land. This is a solid, if unsurprising, paper. The results seem robust and the background well researched. Some of the conclusions over-reach a little. It is well worthy of publication. There are a few minor improvements I would recommend.

The introduction is rather long for a paper with this amount of results, and many of the

points are repeated in the discussion. Ideally, each point should only be made once, and some rationalization of the amount of general background given would be good too.

Reply: We have rewritten the discussion section reducing the amount of repeated information. We now only repeat information where we feel it to be necessary for clarity.

p 2, line 7. Who is the author of this editorial? The original source/s would be better.

Reply: Unfortunately, no information about the authors is available. We replaced the sentence and the reference so it is more relevant for our study now (p 2, line 7-9 in the revised manuscript with marked changes):

"Based on the HYDE data set, Campbell et al. (2008) estimated that 269 Mha of cropland and 479 Mha of pasture have been abandoned between 1700 and 2000."

p 4, line 14 and 20. The amount of N fertilization to crops and how much is 'left behind' seems to be a critical aspect of the story, as N limitation becomes important in the subsequent LUC back to 'natural' (e.g. the discussion on p 7/8, lines 22 - 4). Some context of the size of this (where did the fertilization value come from?), and discussion of this assumption's effects on the results are necessary.

Reply: As we were performing stylized simulations, we did not intend to account for the large spatial variability in N application rates. We chose a value of 75 kg N ha-1 yr-1 as a compromise between higher values in intensively managed croplands in parts of Europe and lower values as presently found in large parts of Africa (e.g. Potter et al. 2010). We added the following text (p 4, line 22-24):

"We simulated N fertilization in croplands by applying 75 kg ha-1 yr-1 equally throughout the year to sustain crop productivity with time. This value represents a compromise between higher values presently found in parts of Europe and lower values in most of Africa (e.g. Potter et al., 2010)."

p 5, line 10. I'm a little unsure about the exclusion of desert and tundra. If all grid-cells above 62.5N are excluded from the results, why are they included in the simulations, or the results plots? Why not just have the map end at 62.5N and save the space?

Reply: This is a good suggestion. We have adjusted the maps accordingly.

With regards to the desert, I'm wondering what (if any?) representation of irrigation there is, and whether if there is irrigation, 'desert greening' with irrigated cropland might be an interesting aspect of this study.

Reply: The reviewer raises an interesting point about desert greening via irrigation. We did not consider irrigation in our simulations. Although irrigation in deserts would clearly have a notable effect, the area of irrigated desert is however small globally.

p 6, line 19. Check the sense of this sentence.

Reply: We restructured the sentence, hopefully it is clearer now (p 6, line 26-27):

"In contrast, recovery times are clearly longer (>100 years) in other parts of the temperate forests and in the tropical forests."

p 7, lines 5 - 8. This ought to be in the figure caption, not the main text.

Reply: We have moved the text to the figure caption.

p 9, lines 1-3. This tropical soil carbon response is interesting - can you enlarge on what the physical or model mechanism that causes it is? What causes the change from soil carbon loss to accumulation?

Reply: The two phase aspect of the soil C response results from the differing effect of cropland on individual soil carbon pools. Only the faster-decaying pools are affected by tillage (see Pugh et al., 2015 for a justification), and loss of carbon from these pools dominates the response of the system in the initial period after conversion. Once these faster-decaying pools have come into balance with the influence of tillage on soil respiration rates, then the slow accumulation of carbon in pools not affected by tillage

comes to dominate the overall response. We added the following sentence to the text (p 9, line 18-20):

"This occurs because tillage-driven C losses in more labile soil pools, which dominate the system's response during the first decades, are eventually supplanted as the dominant process by accumulation in more stable pools."

The discussion is very long, rather dry, and as a reader it is difficult to get a clear overall sense of how well the model results compare to field observations. The simple fix for this would be a table with: observation type (e.g. soil carbon recovery time in pasture); observation value (e.g. 100 years); closest model (e.g. P20); closest model value (e.g. 50 years); model performance (either + (too high), - (too low), or a tick (within the obs. uncertainty)). This would convert about 4 pages of hard-to-digest discussion into 1 page of at-a-glance clear results.

Reply: We agree that the discussion could be more condensed and much of the provided information (which necessarily represent only a subset of all available studies anyway) are not absolutely crucial for the study. We thus added the recommended table (Table 2), although it was often difficult to summarize the main findings and characteristics of the studies and their comparability to our results in just a few words. This, as the reviewer correctly pointed out, allowed us to shorten the original text.

Please could you define acronyms and unusual terms, when they are first used? e.g. swidden, NEE, NBP.

Reply: We removed the sentence containing "swidden" and added the definition to NEE (p 15, line 20-21). NBP was already defined when first used (p 5, line 18-19).

p 13, line 10. I'm not sure it's accurate to dismiss LUC biogeophysics as irrelevant at a global scale. (For where biogeopysics has a significant impact on the global climate, see for instance, Davies-Barnard et al., 2014; Davin and de Noblet-Ducoudré, 2010; Jones et al., 2012; Matthews et al., 2004; Pongratz et al., 2010.)

Reply: We did not intend to claim biophysical effects on global climate to be irrelevant and agree that the impacts are still regarded controversial. We have rephrased the sentence to make this clearer (p 14, line 1-5):

"Substantial impacts related to realistic land-use have been found on local-to-regional scales (Alkama and Cescatti, 2016; Peng et al., 2014). Whether or not the locally observed changes translate to a significant global radiative forcing is still debated as the direction of change differs across regions in some climate models, which may cancel when integrated globally (Pielke et al., 2011)."

p 13, line 22. Ill defined is more the case than subjective.

Reply: Corrected.

p 13, line 24. Check the sense of this sentence.

Reply: We shortened the sentence (p 14, line 15-17):

"By our definition we do not capture situations e.g. where the system approaches towards a new equilibrium (as soil C did in some regions in the cropland simulations)."

p 14, lines 7 - 15. Why is +/- 1 sd not the default way of analysis in this study? Wouldn't that make much more sense?

Reply: We indeed considered whether an ecosystem should be designated as "recovered" in cases where the variable's value was initially higher than under reference conditions. However, while recovery from lower or higher levels points towards different mechanisms, one would not easily be able to distinguish both cases. We decided to study recovery from a depletion perspective (which is what people usually link to this term) and thus only used a lower threshold in our default definition as this is the one threshold that is critical from a depletion perspective. In any case, both options result in very similar recovery times in most biomes (Figure A5).

The conclusions need to be more specific, and restricted to results that can be directly

evidenced from the results in the paper.

For instance, in #5, BVOCs are the main point being made - but BVOCs aren't included in the model or the paper, and the point is referenced elsewhere. If you have to reference another paper in your take home messages, you really should rethink what *your* take home messages are, because currently, they are someone else's.

Reply: We believe that conclusions should not be restricted to the direct findings of a study but should also consider the "bigger picture". BVOCs are just an example for possible implications of our study on subsystems beyond the land; while we were not able to account for BVOCs in this particular study, it could be principally done in LPJ-GUESS. Still, to avoid misinterpretation we removed the BVOC part from the respective sentence.

Conclusion #3 also particularly suffers from lack of evidence. The results here show that for most variables, an equilibrium simulation of 100 years is plenty to sort out any legacy LUC effects. Soil carbon is an exception, but then there is lots of evidence that soil carbon is very uncertain in both models and observations. Longer simulations would do the same as re-growth dynamics in many cases, and so you need to highlight which variables in which regions under what time-scales, you have shown are affected.

Reply: Vegetation C and composition indeed (nearly) recover within 100 years in most ecosystems, but we don't see how extended simulations would solve the problem. While this approach would indeed reduce land-use legacy effects in equilibrium simulations, it is not an option in more common transient climate simulations where the system is never in an equilibrium state. We added the following to conclusion #3 (p 16, line 11-14):

"Our study suggests that for vegetation and soil C studies, accounting for LUC over the last 100-150 years is sufficient in the tropics, while in the temperate and boreal zone more than 200 years might be necessary; studies restricted to vegetation should not have to account for LUC more than 150 years ago in any major climatic zone."

The color schemes, especially on the maps, are not all that easy on the eye, and would be very difficult for someone who is color-blind to interpret. Could you consider another color scheme? You could look to http://colorbrewer2.org/ for some good, easy color schemes.

Reply: To ease matters for color-blind readers we changed the green-to-red color scheme to a blue-to-red one in Figure 3/A3/A4/A5.

Reviewer's references:

Davies-Barnard, T., Valdes, P. J., Singarayer, J. S. and Jones, C. D.: Climatic impacts of land-use change due to crop yield increases and a universal carbon tax from a scenario model, J. Clim., 27(4), 1413–1424, doi:10.1175/JCLI-D-13-00154.1, 2014.

Davin, E. L. and de Noblet-Ducoudré, N.: Climatic Impact of Global-Scale Deforestation: Radiative versus Nonradiative Processes, J. Clim., 23(1), 97–112, doi:10.1175/2009JCLI3102.1, 2010.

Jones, A. D., Collins, W. D., Edmonds, J., Torn, M. S., Janetos, A., Calvin, K. V., Thomson, A., Chini, L. P., Mao, J., Shi, X., Thornton, P., Hurtt, G. C. and Wise, M.: Greenhouse Gas Policy Influences Climate via Direct Effects of Land-Use Change, J. Clim., 26(11), 3657–3670, doi:10.1175/JCLI-D-12-00377.1, 2012.

Matthews, H. D., Weaver, A. J., Meissner, K. J., Gillett, N. P. and Eby, M.: Natural and anthropogenic climate change: incorporating historical land cover change, vegetation dynamics and the global carbon cycle, Clim. Dyn., 22(5), 461–479, doi:10.1007/s00382-004-0392-2, 2004.

Pongratz, J., Reick, C. H., Raddatz, T. and Claussen, M.: Biogeophysical versus biogeochemical climate response to historical anthropogenic land cover change, Geophys. Res. Lett., 37(8), L08702, doi:10.1029/2010GL043010, 2010

---

## Author Comment (AC2) · 23 Jun 2016

We thank the two reviewers for their comments which certainly helped to improve the structure and readability of our manuscript. In the following we answer their comments.

Reviewer #2: This manuscript evaluates ecosystem recovery after disturbance from agriculture and pasture management using a dynamic vegetation model and idealized case studies. This is an interesting application of the LPJ-GUESS model, however the results are not particularly novel. The study was well-executed and the paper is clearly written. I suggest publication with some minor revisions mainly to improve clarity.

General comment: I find the inconsistency in terminology relative to "vegetation composition" to be somewhat confusing. First the term vegetation composition is used (abstract); later it is defined as the LAI of the dominant PFT (page 3, L 15), and then

finally in section 2.4 the reader learns that it is both the dominant PFT and the dominant PFT LAI. Therefore, it's not clear which result is presented in the figures since it seems to flip from LAI in figure 2 and dominant PFT in subsequent figures. It should be obvious to the reader how dominant PFT is defined and kept consistent throughout the manuscript.

Reply: We assumed the dominant PFT to be representative for vegetation composition recovery. As declared in section 2.4, we tested two preconditions to check the recovery of the dominant PFT, the recovery of its LAI compared to the LAI in the reference simulation (analogue to the other variables) and the PFT's dominance compared to other PFTs (which was achieved instantly for grasses and usually earlier than the LAI recovery for woody PFTs). In Figure 2 we only show changes in the LAI of the dominant PFT (precondition 1) because there is no simple way to show dominance recovery (precondition 2) in this figure (which is most often not relevant anyway). We made adjustments in the text to achieve a uniform presentation of this definition and to avoid confusion.

Page 5, L 29-30: reads awkwardly, to what does lower limit refer? Assuming that 'lower limit' implies the time to return to pre-disturbance conditions, wouldn't it be more appropriate to say that the ecosystem might never return. It is possible those sites have reached new equilibriums; does a trend still exist or has the model reached a new steady state?

Reply: Yes, lower limit refers to the recovery time which could theoretically lie between 801 years and infinity (new equilibrium) in these cases. For soil C (the only variable for which significant areas do not recover within 800 years), there is often a small (P100) to very small (C100) trend, suggesting that most of these "black" areas would eventually recover as well within some more centuries. However, the C accumulation rate usually becomes much lower some decades/few centuries after agricultural abandonment. Given the relatively low amount of C stored in these regions and the fact that most of the originally lost C is recovered within one or two centuries the importance of the very

long recovery times in these areas for the global C cycle should not be overrated. We made the following changes to the text for clarity:

"For all variables, the recovery time was capped at 800 years after reconversion to natural vegetation when recovery was not achieved at the end of the simulation period, implying a lower limit for the recovery time. However, the actual recovery time in these cases could theoretically lie between 801 years and infinity." (p 6, line 3-6 in the revised manuscript with marked changes)

"It should be noted that even though some regions do not recover within 800 years, a large fraction of the original C loss is already replenished after few centuries, thereby limiting implications for the C cycle." (p 9, line 8-10)

Page 7, L1-2: This is interesting, looking at Figure 3 – there is a line of grids in the boreal forest that has exceptionally long time to recover LAI right next to grid cells that only take a fraction of that time to recover. The authors mention this, but provide no reason for the behavior. Looking at figure 5, both P100 and C100 have similar levels of N limitation in this region, so why does C100 take so much longer than P100 to recover?

Reply: In the C100 simulation, other PFTs than the dominating one (which in this case is IBS) are more competitive once tree establishment occurs. Thus, while total LAI is similar to the other simulations, the LAI of the dominant PFT is substantially reduced, thereby remaining below the threshold in some regions. We interpret this behavior as the PFT's response to varying soil available N levels. Total N limitation seems indeed similar for P100 and C100 in this region, however, available N is still higher for P100 and even small differences can be important as IBS generally suffers more from N limitation than other PFTs. We added this explanation (p7, line 9-11):

"Furthermore, in parts of the boreal zone recovery takes several hundred years for C100 instead of a few decades for the other simulations because lower available N levels relatively reduce the growth of IBS (the dominant PFT in this region) compared

to other woody PFTs."

Page 7, L14-16: This sentence should be rewritten for clarity.

Reply: We shortened the sentenced to (p7, line 26-28):

"Compared to dominant PFT, recovery occurs slightly later for vegetation C (Fig. 2, Table 1)."

Page 7, L17-18: The fact that grasslands return so quickly isn't surprising given the model setup. Although the authors don't specify how crops are modeled (other than some C and N modifications), I suspect that they are otherwise modeled exactly the same as grasses. If this is the case, grassland recovery would be almost immediate relative to dominant PFT and vegetation carbon.

Reply: Croplands are modeled as pastures (grasslands) apart from the differences described in section 2.1. Quick vegetation and LAI recovery is indeed not surprising, however, if productivity is reduced e.g. due to nitrogen limitation, LAI or vegetation C recovery will sometimes take several decades or more (Figure 3).

Page 7, L21: Boreal forests seem to have a higher standard deviation than tropical in figure 6.

Reply: That's true, but here we compared the standard deviations of dominant PFT (e.g. the black bar with the green background in the PFT figure) to the standard deviations of vegetation C (e.g. the black bar with the green background in the Veg C figure), not standard deviations of one variable for different biomes (which would be e.g. the black bar with the green background in the PFT figure to the black bar with the blue background in the PFT figure).

Page 8, L 6: The soil C loss of 0-11% seems very low, especially for agriculture lands. I would expect a minimum of 20% loss (even for a short 20 year period). This seems to be confirmed in the discussion section.

Reply: The relatively low values in our study could have a number of reasons. While observations are often restricted to the top soil C loss (where changes tend to be more pronounced), the model has no soil depth, meaning that our changes are implicitly averaged over the whole soil profile. Moreover, we excluded litter C from the analysis (mentioned in section 2.4). While soil C is usually larger than litter C, relative depletions are larger for the litter. In addition, our assumed fertilizer rate might be higher than historic fertilizer application, thereby overestimating C input to the soil. We added the following sentences to the discussion (p 10, line 22-25):

"LPJ-GUESS tends to simulate lower C loss in croplands than commonly reported in observations. We attribute this to a combination of the observation's focus on the top soil (while in LPJ-GUESS soil C is implicitly averaged over the whole soil column), the exclusion of litter in our analysis, and our relatively high fertilizer rates."

Page 8-9, L29-31-L1-3: This is unexpected and deserves some explanation.

Reply: We added a sentence about the cause (see also comments to reviewer 1) (p 9, line 18-20):

"This occurs because tillage-driven C losses in more labile soil pools, which dominate the system's response during the first decades, are eventually supplanted as the dominant process by accumulation in more stable pools."

I'm also curious, how is soil C recovering so quickly for P100 and C100 (for example in central Africa) when the other components (LAI, Veg C, NBP) are taking much longer (a hundred years or more).

Reply: It's important to keep in mind that while vegetation C and dominant PFT LAI are generally higher in the tropics than in other forest biomes, soil C is actually lower (see Figure 1), which means more vegetation C and less soil C has to be accumulated to reach background conditions (even though relative depletions are similar). Also, while tropical productivity is much higher than in temperate and boreal forests, biomass

accumulation rates are similar, pointing towards higher turnover rates in tropical forests and possibly also increased mortality. The C input to the soil is thus enhanced and the high soil turnover rates allow soil C to response quickly to the new conditions.

Page 10-13, Section 4.1: This section tends to ramble and focus on elements that are not related to this study. At times, it is difficult to read and should be shortened and focused.

Reply: We added a table (Table 2) with the relevant information while removing large parts of the text (see also comments to reviewer 1).

Page 10, P1: The authors did a poor job of comparing the model output with observations. First, the authors never mention the soil depth of the model, so I don't think any of the comparisons with observations in this section are useful. Second, comparing against other versions of the model that didn't have the N cycle doesn't add any useful information regarding model performance capturing soil C loss after disturbance. I understand that soil C loss isn't the focus of the paper, but if the reader can't trust the soil c loss from management (which I suspect is underestimated), how can they trust the recovery estimates.

Reply: The primary aim of our study was not to reproduce site-specific soil C losses quantitatively but to estimate the legacy effects of different LU histories on the modelled carbon cycle and vegetation regrowth. We did not report soil depth simply because the model has none (we have now made this explicit in the text in section 4.1). Whilst this, and the fact that we did not use site-specific characteristics, indeed makes quantitative comparisons to observations difficult (which we have attempted to state more clearly in the revised manuscript), we still think the model should nevertheless capture the general tendencies. The fact that the model successfully reproduced a range of observations (see later in the discussion) and also captures most of the patterns observed after agricultural abandonments strengthens the reliability of our results. We mentioned the Pugh et al. (2015) study to give an idea how a different model setup

can modify long-term C fluxes in agricultural soils. We extended one sentence in the discussion (p 11, line 1-4):

"By contrast to studies of LU effects compared to previously natural ecosystems, the regeneration of ecosystems after agricultural abandonment has been studied less, and a direct comparison to our simulations is challenging, either because limited information about former LU or reference conditions was provided in these studies, or because there are important differences from our setup in terms of management and LU duration or other site-specific characteristics."

Page 10, L 28-32: Reword for clarity.

Reply: We removed the sentence and put the information into Table 2.

Page 11, L 29-30: This sentence is not clear.

Reply: We removed the sentence and put the information into Table 2.

Conclusion: I think one conclusion that wasn't made (that could be based on the alternate recovery results) is that recovery for some variables doesn't seem to be ever reached, only a new equilibrium (particularly for soil carbon).

Reply: As mentioned before, even though for soil C significant areas do not recover within the simulation period of 800 years, we think that finding should not be over-interpreted. It might take some more centuries or millennia until soil C in these areas would actually be "recovered" according to our definition, but the actual amount of C "missing" to reach reference conditions is usually very small, thereby limiting impacts on the global C cycle. We added the following point to conclusions #5 (p 16, line 21-23):

"5. In terms of soil C, our results suggest that some subtropical regions might not recover at all on timescales relevant for humans. However, given the low absolute amounts of C "missing" in these soils, implications on the global C cycle are expected to be small."

Table 1: caption reads "Recovery times are depicted in Figure 4" – should be Figure 3.

Reply: Corrected.

Table 1: For the dominant PFT LAI recovery, I find it interesting that for a temperate forest, the P100 and C100 take less time to recover than the P20 and C20. Although all times are within the error bars, it still is not consistent with the other biomes. The authors don't mention this behavior in Section 3.2, but it would be nice to have an explanation.

Reply: This pattern is generally found in regions where the TeBS PFT dominates, while regions where TeBE or IBS dominate recover typically faster for P20 and C20. We interpret this behavior as that reduced soil N favors TeBS in the competition with other tree PFTs, thereby reaching its background LAI levels earlier. We added this explanation to the results (p 7, line 22-24):

"Interestingly, temperate forests recover faster for P100 and C100 then for P20 and C20. This pattern is generally restricted to areas where the TeBS PFT dominates. We interpret this behaviour as that reduced soil N favors TeBS in the competition with other tree PFTs, thereby reaching its background LAI levels earlier."

Figure 1 has a lot of acronyms that aren't defined, please define each PFT.

Reply: We added a table (Table A1) for the PFTs.
* * *

---

## Author Response (AR1)

We thank the editor for his additional comments which helped to further improve the clarity of our manuscript. In the following text we answer the comments point-by-point. The editor's comments are written in black, our response in blue and changed text to the manuscript in green.

Comments to the Author:

Dear Authors,

Thank you for your replies to reviewers' comments. I have now read through reviewers' comments as well as your replies. Based on my reading of the paper and your replies, I am happy to have you change your manuscript as you suggest in your replies and submit a revised version of the manuscript. However, based on my own reading of the manuscript, I would request you to take into account the following minor comments mainly to improve the clarity of your manuscript and address some minor issues.

1. I don't think N, B and P in Net Biome Productivity need to be upper case every time you write this term. Similarly for Plant Functional Type.

We changed these terms to lower case where the terms are written out. Abbreviations are kept to NBP, PFT etc.

2. In agreement with reviewer #2, I am confused about how you define the dominant PFT. In my simple mind, a dominant PFT is the one which occupies the largest fraction in a grid cell. Whatever definition you have used, unless I missed it, should be explicitly mentioned in the manuscript. The units of dominant PFT would be fractional coverage. Isn't it? Also, in Figure 3, having read reviewer # 2's comment I am now confused if "Dominant PFT" refers to the fractional coverage of the dominant PFT or LAI of the dominant PFT.

Which PFT dominates in a grid-cell was decided based on LAI. In our study, a PFT can be dominating even though its LAI is not yet similar to its LAI in the reference simulation (and vice versa). In other words, the recovering LAI of the dominant PFT in e.g. the P20 simulation (which PFT is dominant in a grid-cell is derived from the reference simulation) is compared to the LAI of other PFTs in the P20 simulation (precondition 2) and to the LAI of the dominant PFT in the reference simulation (precondition 1). Both preconditions (dominance and LAI recovery) have to be fulfilled before dominant PFT recovery occurs. We hope we made this clearer now in the text by changing some sentences and giving an example (p6, line 1-12 of the revised manuscript):

"For the dominant PFT recovery, we first identified which PFT dominates each grid-cell in the reference simulation based on the annual maximum leaf area index (LAI) amongst PFTs. We then checked for dominant PFT recovery the same way as we did for vegetation C, soil C and NBP (i.e. if its LAI exceeded the threshold of 1 σ below the reference simulation mean; condition 1) but additionally checked if its LAI was also larger than the LAI of any other PFT in the same simulation and year (i.e. the dominant PFT is the same as in the reference simulation, precondition 2). Thus, dominant PFT recovery was only possible if both conditions were fulfilled. For example, if the temperate

broadleaved evergreen (TeBS) tree was the dominant PFT in the reference simulation (with an average maximum LAI of e.g. 3.0 and standard deviation of ± 0.2), dominant PFT recovery in a specific LU simulation (e.g. P20) would occur once the LAI of TeBS in this simulation a) hits the threshold of 2.8 (3.0 - 0.2, condition 1) and b) is larger than the LAI of any other PFT in P20 in the specific year, i.e. TeBS is the dominant PFT in the grid-cell (condition 2). For all variables, the recovery time was capped at 800 years after reconversion to natural vegetation, the point when simulations ended. Recovery times of 800 years thus represent a lower limit. However, the actual recovery time in these cases could theoretically lie between 801 years and infinity."

3. Please add titles to the four panels in Figure 1.

Done.

4. Just to reiterate reviewer #2's comments it was difficult to read the manuscript with no table explaining the acronyms for PFTs.

We added Table A1 which explains the PFT acronyms.

5. Page 4, line 24. Replace "species" by 'PFT". LPJ doesn't represent species, it represents PFTs. Correct?

That's correct but species referred to the observations in this case. We changed "species" to "PFTs" in the previous sentence instead which indeed related to processes within LPJ-GUESS (p4, line 23-25 of the revised manuscript):

"After patch-destroying disturbances or managed land converting back to natural vegetation, there is a typical succession from grasses to light-demanding pioneer trees, eventually followed in many ecosystems by the establishment of shade-tolerant PFTs."

6. Page 4, line 29. Please consider replacing "… including example of all major biomes" with "These regions include all major biomes".

We replaced this part of the sentence. It now reads (p4-5, line 31-1 of the revised manuscript):

"These regions include all major biomes (Smith et al., 2014)."

7. Page 5, line 4. Please consider replacing "a varying time period of 20, 60 and 100 years" with "time periods of 20, 60 and 100 years" here and elsewhere.

We replaced this term everywhere.

8. Page 5, line 14. The manuscript reads, "the sum of all ecosystem-level fluxes, termed Net Biome Productivity (NBP)". This is a vague definition and may not be correct depending on what fluxes are included. Please reword this properly.

We changed the sentence to (p5, line 15-17 of the revised manuscript):

"We studied the influence of LU history on ecosystems by analyzing four key variables: dominant PFT, vegetation C, soil C (excluding litter) and the sum of all ecosystem-level C fluxes (including NPP, soil respiration, fire, harvest, land clearing, and decomposition of the product pool), termed net biome productivity (NBP)."

9. Page 5, lines 20-30. Just like reviewer #2, I am lost here. Although Figure A1 helps this section needs to be reworded for an easy read.

We revised this section and hope it is clearer now (see also #2) (p5-6, line 15-12 of the revised manuscript):

"We studied the influence of LU history on ecosystems by analyzing four key variables: dominant PFT), vegetation C, soil C (excluding litter) and net biome productivity (NBP) (the sum of all ecosystem-level C fluxes including NPP, soil respiration, fire, harvest, land clearing, and decomposition of the product pool). We investigated the legacy effects of LU history by calculating a recovery time for each variable, simulation and grid-cell after the conversion back to natural vegetation. For vegetation C, soil C and NBP recovery time was defined as the year in which the 20-year running mean of the variable exceeded the threshold of one standard deviation ($\sigma$) below the mean of the reference simulation (full simulation period) for the first time after agricultural abandonment. $\sigma$ was calculated on the 20-year running mean of the reference simulation. To avoid "false positive" identifications of recovery in cases for which the variable of interest was initially within 1 $\sigma$, but then exhibited dynamics taking it outside this range (e.g. soil C in Fig. A1), we applied an additional criterion of whether the minimum after the transition to natural vegetation occurred in the first 200 years and if it was below the mean minus 1 $\sigma$ threshold. If that was the case, the condition was expanded so that the variable could only be defined as recovered after the year in which the minimum occurred ("minimum rule"). A 200 year window was chosen because the minimum occurred within the first 200 years for all biome averages of all variables and simulations. If the minimum was located after 200 years, we assumed the minimum to be a result of natural variability and recovery was achieved as soon as the variable in question exceeded the threshold of 1 $\sigma$ below the reference mean. Figure A1 shows an example of how soil C recovery was calculated for one site.

For the dominant PFT recovery, we first identified which PFT dominates each grid-cell in the reference simulation based on the annual maximum leaf area index (LAI) amongst PFTs. We then checked for dominant PFT recovery the same way as we did for vegetation C, soil C and NBP (i.e. if its LAI exceeded the threshold of 1 $\sigma$ below the reference simulation mean; condition 1) but additionally checked if its LAI was also larger than the LAI of any other PFT in the same simulation and year (i.e. the dominant PFT is the same as in the reference simulation, precondition 2). Thus, dominant PFT recovery was only possible if both conditions were fulfilled. For example, if the temperate broadleaved evergreen (TeBS) tree was the dominant PFT in the reference simulation (with an

average maximum LAI of e.g. 3.0 and standard deviation of ± 0.2), dominant PFT recovery in a specific LU simulation (e.g. P20) would occur once the LAI of TeBS in this simulation a) hits the threshold of 2.8 (3.0 - 0.2, condition 1) and b) is larger than the LAI of any other PFT in P20 in the specific year, i.e. TeBS is the dominant PFT in the grid-cell (condition 2). For all variables, the recovery time was capped at 800 years after reconversion to natural vegetation, the point when simulations ended. Recovery times of 800 years thus represent a lower limit. However, the actual recovery time in these cases could theoretically lie between 801 years and infinity."

10. Page 6, line 3. Please replace "Maps of vegetation and soil C …" with "Maps of simulated vegetation and soil C …"

We changed this part of the sentence accordingly (p6, line 16-17 of the revised manuscript):

"Maps of simulated vegetation and soil C, as well as dominant PFT and biomes derived from PFT composition for the reference simulation are shown in Fig. 1."

11. Page 6, lines 22-25. These sentences about pre-condition 1 and 2 are not clear at all to me.

Both preconditions have to be fulfilled for recovery to occur. If one precondition (e.g. precondition 2) is fulfilled but the other (precondition 1) isn't, recovery is delayed and precondition 1 would be the delaying precondition in this case. With the rewritten methods section we hope this is clearer now.

12. Figure 6. It would be really useful to label panels as a), b), c) and d) and refer to the panels in text.

Done.

13. Page 6, line 2 reads "Figure 6 shows the maximum differences in recovery times …". Differences between what and what? Aren't these just the recovery times? Here and elsewhere you have used the word "difference" and I can't appreciate what's the difference between?

No, it's the maximum differences between recovery times across simulations. We added the following example to the figure to make this clearer (p30 of the revised manuscript):

"For example, if recovery times for one variable in one grid-cell would be 50, 60, 65, 90, 100, 110 years (for P20, P60, P100, C20, C60, C100), the maximum difference in recovery time across all simulations (black) would be 60 years, across the 20 year simulations (green) 40 years, across the 60 year simulations (blue) 40 years, across the 100 year simulations (red) 45 years, across the pasture simulations (orange) 15 years and across the cropland simulations (purple) 20 years."

14. Page 7, line 7. Shouldn't the "LU type" be "LU duration" here.

No. We changed the sentence (now moved to label Figure 6) to (p30 of the revised manuscript):

"Thus, the black dots show the sensitivity of recovery times to LU history across all simulations for each biome. The red, blue and green squares indicate the relative contribution of LU type for a specific LU duration to this sensitivity, and the orange and purple squares the relative contributions of pasture and of cropland duration."

15. Page 7, line 8 reads "While substantial differences occur across the pasture simulations ...". DO you mean differences across P20, P60 and P100.

Yes. We changed the sentence to (p7 line 20-24 of the revised manuscript):

"While substantial differences occur across the pasture simulations (P20, P60, P100) in tropical forests, savannas and grasslands, and across cropland simulations (C20, C60, C100) in boreal forests (emphasising the importance of LU duration in these regions), major differences between P100 and C100 occur in boreal forests and grasslands (emphasising the importance of LU type if agricultural duration was long)."

16. Page 7, line 8 reads " ... and across croplands in boreal forests ...". Do you mean "across cropland simulations (C20, C60 and C100)".

Yes. We changed the sentence (see comment #15) to (p7 line 20-24 of the revised manuscript):

"While substantial differences occur across the pasture simulations (P20, P60, P100) in tropical forests, savannas and grasslands, and across cropland simulations (C20, C60, C100) in boreal forests (emphasising the importance of LU duration in these regions), major differences between P100 and C100 occur in boreal forests and grasslands (emphasising the importance of LU type if agricultural duration was long)."

17. Page 7, lines 11 and 12 read "On the other hand, in our simulations dominant PFT recovery in temperate forests is hardly influenced by the type of former LU, or, conversely, pasture duration has negligible effects on boreal forest recovery". In the absence of clear definition of "dominant PFT" I can't appreciate if it's the LAI or the fractional coverage of dominant PFT that is being talked about.

Dominant PFT recovery is calculated based on two preconditions (see comment #2), the dominance of the PFT compared to other PFTs (in terms of LAI) and the recovery of the LAI itself. So dominant PFT recovery is based on its LAI but two different preconditions have to be fulfilled. We changed the text about the recovery definition accordingly, including an example for dominant PFT recovery (see also comment #2) (p6, line 1-12 of the revised manuscript):

"For the dominant PFT recovery, we first identified which PFT dominates each grid-cell in the reference simulation based on the annual maximum leaf area index (LAI) amongst PFTs. We then checked for dominant PFT recovery the same way as we did for vegetation C, soil C and NBP (i.e. if its LAI exceeded the threshold of 1 σ below the reference simulation mean; condition 1) but additionally checked if its LAI was also larger than the LAI of any other PFT in the same simulation and year (i.e. the dominant PFT is the same as in the reference simulation, precondition 2). Thus, dominant PFT

recovery was only possible if both conditions were fulfilled. For example, if the temperate broadleaved evergreen (TeBS) tree was the dominant PFT in the reference simulation (with an average maximum LAI of e.g. 3.0 and standard deviation of ± 0.2), dominant PFT recovery in a specific LU simulation (e.g. P20) would occur once the LAI of TeBS in this simulation a) hits the threshold of 2.8 (3.0 - 0.2, condition 1) and b) is larger than the LAI of any other PFT in P20 in the specific year, i.e. TeBS is the dominant PFT in the grid-cell (condition 2). For all variables, the recovery time was capped at 800 years after reconversion to natural vegetation, the point when simulations ended. Recovery times of 800 years thus represent a lower limit. However, the actual recovery time in these cases could theoretically lie between 801 years and infinity."

18. Page 7, lines 14-15 read "Compared to dominant PFT LAI, the starting point of vegetation C recovery averaged over all grid-cells is lower (11-14 % of the reference simulation for vegetation C compared to 39-47 % for dominant PFT) due to higher percentage loss of vegetation C during the period of agriculture and recovery occurs slightly later". Do you mean starting point after abandonment? I also do not follow what "39-47% for dominant PFT" means. Please reword this sentence.

We removed these parts from the sentence. The revised text now reads (p7, line 30 of the revised manuscript):

Compared to dominant PFT recovery occurs slightly later for vegetation C (Fig. 2, Table 1).

19. Page 7, lines 18-20 read "Lower standard deviations for the mean differences in recovery times for most biomes (Fig. 6) reflect the more uniform response of post-agricultural vegetation C accumulation across different sites compared to dominant PFT recovery." Lower compared to what? What does uniform refers to here – uniform in space or across simulations?

We changed the sentence to (p7-8, line 32-3 of the revised manuscript):

"Lower standard deviations for the mean differences in vegetation C recovery times compared to the standard deviations for the mean differences in dominant PFT recovery times for most biomes (Fig. 6) reflect the spatially more uniform response of vegetation C recovery."

20. Page 7, line 27 reads "We interpret this net N flux to the ecosystem as originating from high levels of water stress in these savannas and grasslands …". Isn't it more appropriate to say "This net N flux is partly caused by high levels of water stress …".

We changed the sentence to (p8, line 10-12 of the revised manuscript):

"This net N flux can partially be explained by high levels of water stress in these savannas and grasslands, resulting in greater C and N allocation to roots relative to leaves and thereby decreased harvest removal in this region (Fig. A3)."

21. Page 7, line 21. Just like reviewer #2, I also thought it should have been "boreal forests and grasslands". You make it clear in your reply, so please make sure to clarify this in your revised manuscript as well.

We changed the sentence to (p8, line 3-4 of the revised manuscript):

"Exceptions are tropical forests and grasslands, where the standard deviation is higher for vegetation C recovery compared to dominant PFT recovery."

22. Page 8, line 6 reads "Relative depletion of soil C content under crop and pasture LU is not as large (loss of 0-11 % compared to the reference simulation) as for vegetation C." This is really low compared to observations (e.g. see Wei, X., Shao, M., Gale, W., and Li, L.: Global pattern of soil carbon losses due to the conversion of forests to agricultural land, Scientific Reports, 4, 4062, doi:10.1038/srep04062, 2014) which report a soil carbon loss of around 40%. This seems to be a limitation of LPJ, as reviewer #2 also points out, and must be noted as such.

This should be clearer now as we added the following text to the manuscript (p10, line 15-18 of the revised manuscript):

"LPJ-GUESS tends to simulate lower C loss in croplands than commonly reported in observations. We attribute this to a combination of the observation's focus on the top soil (while in LPJ-GUESS soil C is implicitly averaged over the whole soil column) and our relatively high fertilizer rates increasing productivity and thereby C input to the soil."

23. Page 8, line 1 reads "Upon re-conversion, soil C accumulation is delayed for the pasture simulations …". Delayed compared to what?

We changed the sentence to (p8, line 25-27 of the revised manuscript):

"Upon re-conversion, soil C accumulation is delayed for the pasture simulations compared to the cropland simulations, especially for P20 where the residual roots and other litter left after the original deforestation event continue to decay and soil C decreases for some decades."

24. Page 8, lines 12-13 read "The general delay for pastures is associated with larger heterotrophic respiration rates (not shown) compared to rates calculated in recovering croplands and low litter input in the early stage of regrowth in forested ecosystems." But the low litter input as forest regrow would occur regardless of previous LU type (pasture or croplands) so why is this a factor in differing behaviour of pastures versus cropland abandonment.

Yes but if inputs were high directly after abandonment, we might not see the C loss ("delay") during the first years of regrowth (which is indeed the case for grasslands). As both former pastures and croplands are characterized by low input during the early stage of forest regrowth, we removed this part from the sentence to avoid confusion (p8, line 27-28 of the revised manuscript):

"The general delay for pastures is associated with larger heterotrophic respiration rates (not shown) compared to rates calculated in recovering croplands."

25. Page 8, lines 17-19 read "Additionally, in the intensive LU simulations (P100, C60, C100), vegetation productivity in the boreal region is further reduced compared to the reference simulation in the first 200 years of regrowth (not shown), reducing litter input to the soil even further." Why does this happen?

We think this happens due to N limitation. We changed the sentence to (p9, line 1-3 of the revised manuscript):

"Additionally, in the intensive LU simulations (P100, C60, C100), vegetation productivity in the boreal region is further reduced compared to the reference simulation in the first 200 years of regrowth (not shown) due to N limitation (Smith et al., 2014), reducing litter input to the soil even further."

26. Page 9, lines 6-7 read "the sensitivity of grasslands is mainly due to differences between simulations of different LU type". I think, I know what is meant here but I am not sure. Please reword this part of the sentence.

We changed the sentence to (p9, line 24-27 of the revised manuscript):

"The maximum differences across all simulations (P20/P60/P100/C20/C60/C100) in boreal forests are mainly due to differences across simulations of same LU type but different duration (e.g. P20/P60/P100), whereas the sensitivity of grasslands mainly reflects differences across simulations of different LU type but same duration (e.g. P100/C100), emphasizing the importance of duration and type of agriculture in a range of biomes."

27. Page 9, lines 11-12 read "The sink capacity of the re-growing vegetation is greatest during the first decades and then gradually goes back to the NBP of the reference simulation." This sentence is mis-leading because NBP is a measure of whole ecosystem not just vegetation

We changed the sentence to (p9, line 30-31 of the revised manuscript):

 "The sink capacity of the recovering ecosystem is greatest during the first decades and then gradually returns to the NBP levels of the reference simulation."

28. Page 9, lines 12-13 read "P20 and, to a lesser extent, C20, act as a smaller sink than the other simulations". Act as a smaller sink than what and over what time period?

As written in the text compared to the other simulations. We added the time period to the sentence (the difference gradually decreases but after 100 years differences are relatively small) (p9, line 31-32 of the revised manuscript):

"P20 and, to a lesser extent, C20, act as a smaller sink than the other simulations at least during the first 100 years of regrowth."

29. Page 10, line 28 reads "The sequestration rate of above-ground biomass in regenerating tropical forests …". This sentence reads a bit weird. I think what you mean is "The rate of increase of above-ground biomass …"

We removed this sentence and put the information into Table 2 ("vegetation recovery rate").

30. Page 10, line 31 reads " … even though in the latter case biomass started to evolve only after around two decades." What does "evolves" means here – increase or decrease?

We removed this sentence and put the information into Table 2 (we removed the specific part about evolving vegetation).

31. Page 10, line 32 reads "LPJ-GUESS shows a reduction of vegetation C accumulation with time for both biomes …". This sentence may be better worded as " … a reduction in the rate of increase of vegetation C …"

We removed this sentence and put the information into Table 2.

32. Page 12, line 16 reads " … recovery can occur at similar or even faster speed in the (sub)tropics." Does "(sub)tropics" means "tropics and sub-tropics"?

Yes. We changed the sentence to (p11, line 16-18 of the revised manuscript):

"The impact of LU duration was rarely studied, however, our results suggest that even though longer agricultural durations mostly result in greater initial soil C depletions, recovery can occur at similar or even faster speed in the sub-tropics and tropics."

33. Please consider using "idealized" in place of "stylized" to refer to your experiments.

Done.

34. Page 13, lines 22-23 read "The term recovery is very subjective and, in the absence of a universal definition amongst ecologists, different approaches imply the potential to significantly modify absolute recovery times. By our definition we examine recovery from a C sequestration perspective …". Please consider modifying this sentence as "The term recovery is subjective and, in the absence of a universal definition amongst ecologists, several definitions currently exist. The definition used in this study examines recovery from a C sequestration perspective …"

We changed the sentence to (p12, line 24-26 of the revised manuscript): "The term recovery is subjective and, in the absence of a universal definition amongst ecologists, several definitions currently exist. The definition used in this study examines recovery from a C sequestration perspective which does not capture situations e.g. where the system approaches a new equilibrium (as soil C did in some regions in the cropland simulations)."

35. Page 14, lines7-9. Please rewrite these sentences to make your point more clear.

We changed the sentence to (p13, line 7-9 of the revised manuscript):

"By expanding our standard recovery definition by an upper threshold (reference mean plus 1 σ), and with the "minimum rule" also applied to the maximum (see section 2.4), one can test if some ecosystems recover from higher rather than lower values than in the reference simulation. Mostly grasslands are affected by this alternative definition (Fig. A5)."

36. Page 14, line 10. The sentence "Mostly grasslands are affected." Seems incomplete.

We changed the sentence to (p13, line 9 of the revised manuscript):

"Mostly grasslands are affected by this alternative definition (Fig. A5)."

37. Page 14, lines 11-12 read "Patterns are similar for vegetation C but more pronounced, especially for croplands." What does "pronounced" means here – higher or lower?

We changed the sentence to (p13, line 11-13 of the revised manuscript):

"Patterns are similar for vegetation C but the increase in vegetation C recovery times is often larger than the increase in dominant PFT recovery times, especially for croplands."

38. Page 14, lines 13-15 read "We do not use an upper limit in our primary definition however, because, in the case of C storage, the ecosystem is already operating at a level of service above that which the unmodified ecosystem would have provided." Upper limit of what? Also, this sentence is not completely clear so please reword it.

We changed the sentence to (p13, line 14-16 of the revised manuscript):

"However, we do not use an upper threshold in the primary definition used in this study because, in the case of C storage, the ecosystem is already operating at a level of service above that which the undisturbed ecosystem would have provided."

39. Page 14, line 18 reads "Elements of random fluctuations due to natural variability made a clear identification of recovery period difficult in that case." Where are these "elements of random natural

variability" coming from if you are driving your model offline with climate data that is being repeatedly used.

While we indeed used a repeating climate cycle, there is additional variability in the model arising from stochastic processes and disturbances and responding C, N, and water dynamics. We added this to the text (p13, line 19-20 of the revised manuscript):

 "Elements of random fluctuations due to natural variability arising from stochastic processes and disturbances and responding C, N, and water dynamics made a clear identification of recovery period difficult in that case."

40. Page 14, lines 28-29 read "This is particularly relevant for flux tower measurements where ongoing regeneration might be overlooked due to large variability of NEE." What does "regeneration" means here? Do you mean inter-annual variability in NEE makes it difficult to see the underlying long-term trend? If yes, please mention it explicitly.

We changed the sentence to (p13, line 30-32 of the revised manuscript):

"This is particularly relevant for flux tower measurements where an underlying long-term trend caused by the recovery from previous, often unquantified or unknown land-use change, might be overlooked due to a large inter-annual variability of net ecosystem exchange (NEE)."

41. Page 15, line 19. Please replace "climate change and/or atmospheric CO2" with "climate change and/or increasing atmospheric CO2".

Done.

42. The discussion around Figures 4 and 5 will benefit from an additional panel in these figures for the reference simulation.

Done.

43. In Figure 6, please replace "TrFo, TeFo and BoFo" by full biome names.

Done.

[revised manuscript text omitted]

---

## Author Response (AR2)

We again thank the editor for his additional comments which helped to further improve the clarity of our manuscript. In the following text we answer the comments point-by-point. As before, the editor's comments are written in black, our response in blue and changed text to the manuscript in green.

Comments to the Author:

Dear Authors,

Thanks for taking into account reviewers' comments and minor edits suggested by me. The manuscript reads fairly well now. I have read through the revised manuscript again and I am sorry but there are still some minor edits I would like you to address.

Page 2, line 11, please consider removing the word "significantly"

We removed the word.

Page 2, line 24, please change "Analyses of the long-term effects of historical LU are often limited by the unavailability of adequate LU information, or the absence of undisturbed ecosystems, and…" to "Analyses of the long-term effects of historical LU are often limited by the availability of adequate LU information, the absence of undisturbed ecosystems, and…"

We changed the sentence accordingly.

Page 3, line 23. At the end of the sentence ending with "growth strategies" please include "(see Table A1 for PFT acronyms and names)". While you did include Table A1 it is never referred to in the main text of the manuscript.

We added this part to the sentence.

Page 5, line 16-17. The manuscript reads "… and net biome productivity (NBP) (the sum of all ecosystem-level C fluxes including NPP, soil respiration, fire, harvest, land clearing, and decomposition of the product pool)."

Since the above definition of NBP fails to mention the sign convention it is essentially incorrect. Clearly, NBP is not equal to NPP + soil respiration + other terms as your sentence seems to imply. In addition, fire, harvest and land clearing are not fluxes but rather processes. I suggest a more generalized definition below which you may modify to reflect your framework.

Net biome productivity (NBP) is the net atmosphere-land carbon flux after carbon losses associated with respiratory fluxes, fire, harvest, land clearing and decomposition of land use change product pools are subtracted from gross primary productivity.

We agree that the suggested definition is clearer. We changed the sentence to (p.5, line 16-18):

"NBP is the net atmosphere-land carbon flux after C losses associated with respiratory fluxes, fire, harvest, land clearing and decomposition of LUC product pools are subtracted from gross primary productivity."

Page 10, line 21-22. The sentence starting with "For croplands, in their study explicitly represented by a number of managed but unfertilized crop functional types, …" doesn't read right. Please reword this sentence.

We changed the sentence and broke it into two parts, hoping it reads clearer now (p.10, line 21-24):

"Croplands were explicitly represented by a number of managed, but unfertilized, crop functional types in Pugh et al. (2015). They found soil C reductions in Europe and Africa of ~50 % after 100 years of cultivation whereas in our study C losses were much smaller (~12 %), possibly partly due to different tillage effects in the two soil models applied."

Page 11, line 16. Please include "After abandonment" at the beginning of the sentence so that it reads "After abandonment, croplands accumulate C faster than pastures, and recovery …". I think, that's what is implied here.

That's right. We changed the sentence accordingly.

Page 13, lines 4-5. Please consider modifying this sentence as follows. "Vegetation C shows similar patterns to the dominant PFT, however, the differences BETWEEN SIMULATIONS (P20, P60 AND P100) are more pronounced."

Here, we compared the recovery times from the alternative recovery definition to the ones from our standard definition for one specific simulation. We changed the sentence to:

"Vegetation C shows similar patterns to the dominant PFT, however, the differences to our standard definition are more pronounced."

Page 13, lines 13-14. Please consider modifying this sentence

"NBP differences look similar to soil C but the effects are much smaller" as follows

"Differences in the recovery times of NBP look similar to soil C but the effects are much smaller."

This is what is meant here, I presume. Also, what does "effects" refer to here. What kind of effects?

Yes. We wanted to say that spatial patterns of changes in NBP recovery times look similar to soil C but as the absolute changes are so small (that's what was meant by effects) we changed the sentence to:

"The recovery times of NBP are hardly affected."

The following sentence after this on Page 13 reads ...

"However, we do not use an upper threshold in the primary definition used in this study because, in the case of C storage, the ecosystem is already operating at a level of service above that which the undisturbed ecosystem would have provided."

In this above sentence the following text "the ecosystem is already operating at a level of service above that which the undisturbed ecosystem would have provided" is somewhat unclear.

Do you mean that if mean + one standard deviation threshold is used for defining recovery then a given variable will have to reach a value higher than it's mean during the undisturbed stage for recovery to occur?

No, an upper threshold would mean the variable has to decrease to its undisturbed value if it is enriched after the agricultural period. However, recovery from higher values is a different issue than recovery from lower values and in our study we wanted to analyze recovery from a depletion perspective. We added this motivation to the sentence:

"However, we do not use an upper threshold in the primary definition used in this study because in this case the ecosystem is already operating at a level of service above that which the undisturbed ecosystem would have provided and our aim here was to investigate recovery from a depletion perspective."

Lastly, please remove bullet number 6 in the conclusions section. This is not an implication based on your study per se but rather a more general statement.

We removed this bullet number.

[revised manuscript text omitted]